# GUARANTEED OUT-OF-DISTRIBUTION DETECTION WITH DIVERSE AUXILIARY SET

## ABSTRACT

Out-of-distribution (OOD) detection is crucial for ensuring reliable deployment of machine learning models in real-world scenarios. Recent advancements leverage auxiliary outliers to represent the unknown OOD data to regularize model during training, showing promising performance. However, detectors face challenges in effectively identifying OOD data that deviate significantly from the distribution of the auxiliary outliers, limiting their generalization capacity. In this work, we thoroughly examine this problem from the generalization perspective and demonstrate that a more diverse set of auxiliary outliers improves OOD detection. Constrained by limited access to auxiliary outliers and the high cost of data collection, we propose Provable Mixup Outlier (ProMix), a simple yet practical approach that utilizes mixup to enhance auxiliary outlier diversity. By training with these diverse outliers, our method achieves superior OOD detection. We also provide insightful theoretical analysis to verify that our method achieves better performance than prior works. Furthermore, we evaluate ProMix on standard benchmarks and demonstrate significant relative improvements of 14.2% and 31.5% (FPR95) on CIFAR-10 and CIFAR-100, respectively, compared to state-of-the-art methods. Our findings emphasize the importance of incorporating diverse auxiliary outliers during training and highlight ProMix as a promising solution to enhance model security in real-world applications. Compared with other methods, the proposed method achieves excellent performance on different metrics in almost all datasets.

## 1 INTRODUCTION

The OOD problem occurs when machine learning models encounter data that differs from the distribution of training data. In such scenarios, models may make incorrect predictions, leading to safety-critical issues in real-world applications, e.g., autonomous driving (Geiger et al., 2012) and medical diagnosis (Liang et al., 2022). To ensure the reliability of models' predictions, it is essential not only to achieve good performance on in-distribution (ID) samples, but also to detect potential OOD samples, thus avoiding making erroneous decisions in test. Therefore, OOD detection has become a critical challenge for the secure deployment of machine learning models (Amodei et al., 2016; Dietterich, 2017; Li et al., 2023; Liu et al., 2023).

Several significant studies (Hendrycks & Gimpel, 2017; Lee et al., 2018b; Liang et al., 2018b) focus on detecting OOD examples using only ID data in training. However, due to a lack of supervision information from unknown OOD data, it is difficult for these methods for satisfactory performance in detecting OOD samples. Recent methods (Hendrycks & andThomas G. Dietterich, 2019; Wang et al., 2021a; Chen et al., 2021; Ming et al., 2022) introduce auxiliary outlier data during the training process to regularize models. While these methods have exhibited significant improvement compared to previous methods without auxiliary outliers, there remains a notable risk that models may encounter OOD instances that deviate significantly from the auxiliary outliers distribution. As depicted in Figure 1(a)-(b), although the auxiliary outliers enhance the model in detecting OOD, its generalization ability is quite unpromising. The above limitation motivates the following important yet under-explored question: *how to guarantee an effective utilization of auxiliary outliers?*

In this work, we theoretically investigate this crucial question from the perspective of generalization ability (Bartlett & Mendelson, 2002). Specifically, we first conduct a theoretical analysis to demonstrate how the distribution shift between auxiliary outlier training set and test OOD data af-

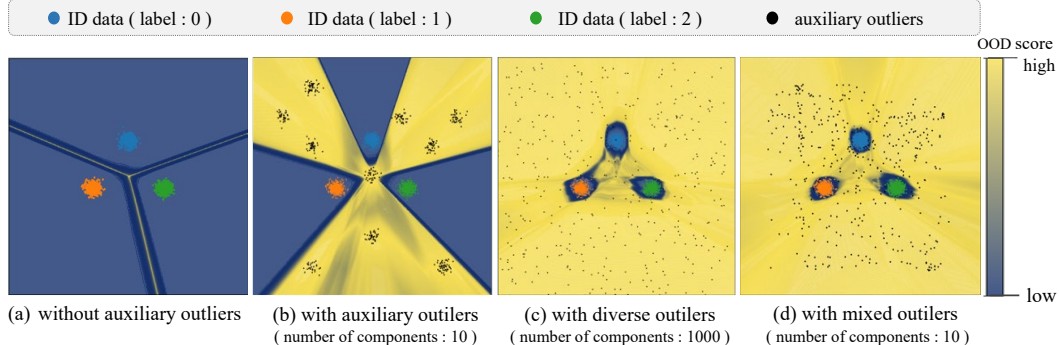

Figure 1: OOD score measurement for different training strategies. The ID data $\mathcal{X}_{in} \subset \mathbb{R}^2$ is sampled from three distinct Gaussian distributions, each representing a different class. The auxiliary outliers are sampled from a Gaussian mixture model away from the ID data, where the number of mixture components indicates the number of classes contained in the auxiliary outliers dataset. (a) The model trained without auxiliary outliers fails to detect OOD. (b) Incorporating auxiliary outliers (10 classes) during training enables partial OOD detection, but overfits auxiliary outliers. (c) OOD detection improves with a more diverse set of auxiliary outliers (1000 classes). (d) Mixup boosts the diversity of auxiliary outliers (10 classes), thereby improving the model's performance.

fects the generalization capability of OOD detector. Then, a generalization bound is provided on the classifier's test-time OOD detection error, considering both its empirical error and the error caused by the distribution shift between test OOD data and auxiliary outliers. Based on the generalization bound, we deduce an intuitive conclusion that *a more diverse set of auxiliary outliers can reduce the distribution shift error and effectively lower the upper bound of the OOD detection error.* As shown in Figure 1(b)-(c), the model trained with a more diverse set of auxiliary outliers achieves better OOD detection. However, in practice, collecting more auxiliary outliers is expensive and the auxiliary outliers we can use are limited. Therefore, a natural question arises - *how to guarantee effective utilization of a **fixed set** of auxiliary outliers?*

An intuitive idea is to enhance the diversity of auxiliary outliers through data augmentation techniques. Accordingly, we propose a simple yet effective method called Provable Mixup Outliers, which introduces the mixup strategy to enhance the diversity of outliers. Mixup employs semantic-level interpolation to generate distinct mixed samples, generating outliers that significantly differ from the originals. This effectively boosts the diversity of auxiliary outliers, as shown in Figure 1(b)-(d), leading to improved OOD detection performance. The contributions of this paper are summarized as follows:

- We provide a theoretical analysis of the generalization error linked to methods trained with auxiliary outliers. By establishing an upper bound for expected error, we reveal the connection between auxiliary outlier diversity and the upper bound of OOD detection error. Our theoretical insights emphasize the importance of leveraging diverse auxiliary outliers to enhance the generalization capacity of the OOD detector.

- Constrained by limited access to auxiliary outliers and the high cost of data collection, we introduce ProMix, a simple yet effective data augmentation method that increases the diversity of auxiliary outliers and is theoretically guaranteed to improve OOD detection performance.

- We observe that ProMix achieves state-of-the-art OOD detection performance, outperforming all compared methods training with auxiliary outliers. Our evaluation on six OOD datasets in standard benchmarks demonstrates significant improvement of 14.2% and 31.5% (FPR95) on CIFAR-10 and CIFAR-100 datasets, respectively.

## 2    RELATED WORKS

In this section, we review the relevant prior research to our work and subsequently conduct a comparative analysis of our approach.

**OOD Detection without Auxiliary Outliers.** Early work by Hendrycks & Gimpel (2017) pioneered the field of OOD detection, introducing a baseline method based on maximum softmax probability. However, it has since been established, as noted by Morteza & Li (2022), that this approach is unsuitable for OOD detection. To address this, various methods have been developed that operate in the logit space to enhance OOD detection. These include ODIN (Liang et al., 2018b), energy score (Wang et al., 2021a; Lin et al., 2021; Wang et al., 2021b), ReAct (Sun et al., 2021), logit normalization (Wei et al., 2022), Mahalanobis distance (Lee et al., 2018b), and KNN-based scoring (Sun et al., 2022). However, post-hoc OOD detection methods that don't involve pre-training on a substantial dataset generally exhibit poorer performance compared to methods that leverage auxiliary datasets for model regularization (Fort et al., 2021).

**OOD Detection with Auxiliary Outliers.** Recent advancements in OOD detection have focused on incorporating auxiliary outliers into the model regularization process. Such as outlier exposure (Hendrycks & andThomas G. Dietterich, 2019) encourages giving predictions with uniform distribution for outliers and Energy-bounded learning (Wang et al., 2021a) widens the energy gap between ID and OOD samples. Prior studies (Wang et al., 2021a; Sehwag et al., 2021; Salehi et al., 2022; Wei et al., 2022) demonstrate improved OOD detection when including outliers during training. Lee et al. (2018a); Grcić et al. (2020); Kong & Ramanan (2021) use properly trained generative models to generate outliers. However, performance heavily depends on outlier quality. ATOM (Chen et al., 2021) uses greedy sampling strategies to select informative outliers to tighten the decision boundary, while POEM (Ming et al., 2022) employs Thompson sampling. Nevertheless, these methods assume access to diverse outliers, which may not always be feasible in practical scenarios.

**Comparison with Existing Methods.** Our approach, ProMix, focuses on enhancing the quality of auxiliary outliers used during training while providing theoretical guarantees. Notably, the most relevant prior algorithms to ProMix are ATOM (Chen et al., 2021) and POEM (Ming et al., 2022), which both recognized that original auxiliary outliers may lack informativeness and proposed methods to mine informative outliers for improving performance. In contrast, our work emphasizes the theoretical significance of auxiliary outlier diversity and enhances outlier diversity through mixup. Importantly, our approach is complementary to ATOM (Chen et al., 2021) and POEM (Ming et al., 2022), as the joint consideration of informative and diverse outliers yields synergistic benefits, resulting in enhanced model regularization.

## 3 THEORY: DIVERSE AUXILIARY OUTLIERS BOOST OOD DETECTION

In this section, we lay the foundation for our analysis of OOD detection. We begin by introducing key notations for OOD detection in Sec. 3.1. Following this, in Sec. 3.2, we establish a generalization bound that highlights the critical role played by auxiliary outliers in influencing the generalization capacity of OOD detection methods. Finally, in Sec. 3.3, we demonstrate how a more diverse set of auxiliary outliers serves to effectively mitigate distribution shift errors, consequently reducing the upper bound of error. For detailed proofs, please refer to *Appendix A*, while a concise summary of our findings is provided below.

### 3.1 PRELIMINARIES

We consider multi-class classification and each sample in the training set $\mathcal{D}_{id} = \{(x_i, y_i)\}_{i=1}^N$ is drawn i.i.d. from the joint distribution $\mathcal{P}_{\mathcal{X}_{id} \times \mathcal{Y}_{id}}$, where $\mathcal{X}_{id}$ denotes the input space of ID data, and $\mathcal{Y}_{id} = \{1, 2, \ldots, K\}$ represents the label space. OOD detection can be formulated as a binary classification problem to learn a hypothesis $h$ from hypothesis space $\mathcal{H} \subset \{h : \mathcal{X} \to \{0, 1\}\}$ such that $h$ outputs 0 for any $x \in \mathcal{X}_{id}$ and 1 for any $x \in \mathcal{X}_{ood}$, where $\mathcal{X}_{ood}$ represents the input space of OOD data with semantics outside the support of $\mathcal{Y}_{id}$ and $\mathcal{X} = \mathcal{X}_{id} \cup \mathcal{X}_{ood}$ represents the entire input space in the open-world setting. To address the challenge posed by the unknown and arbitrariness of OOD distribution $\mathcal{P}_{\mathcal{X}_{ood}}$, we leverage an auxiliary dataset $\mathcal{D}_{aux}$ drawn from the distribution $\mathcal{P}_{\mathcal{X}_{aux}}$ to serve as partial OOD data, where $\mathcal{X}_{aux} \subset \mathcal{X}_{ood}$. Due to the diversity of real-world OOD data, auxiliary outliers cannot fully represent all OOD data, so $\mathcal{P}_{\mathcal{X}_{aux}} \neq \mathcal{P}_{\mathcal{X}_{ood}}$. We aim to train a model on data sampled from $\mathcal{P}_{\widetilde{\mathcal{X}}} = k_{train}\mathcal{P}_{\mathcal{X}_{id}} + (1 - k_{train})\mathcal{P}_{\mathcal{X}_{aux}}$ to obtain a reliable hypothesis $h$ that can effectively generalize to the unknown test-time distribution $\mathcal{P}_{\mathcal{X}} = k_{test}\mathcal{P}_{\mathcal{X}_{id}} + (1 - k_{test})\mathcal{P}_{\mathcal{X}_{ood}}$, where $k_{train}$ and $k_{test}$ determine the proportion of ID data and OOD data used for training and testing, respectively. Note that $k_{test}$ is unknown due to unpredictable test data distribution.

## 3.2 Generalization Error Bound in OOD Detection

**Basic Setting:** We define an OOD label function[1] as $f : \mathcal{X} \to [0, 1]$. The probability that a hypothesis $h$ disagrees with $f$ with respect to a distribution $\mathcal{P}$ is defined as:

$$\epsilon_{\mathcal{P}}(h, f) = E_{x \sim \mathcal{P}}[|h(x) - f(x)|]. \tag{1}$$

Additionally, we define the set of ideal hypotheses on the training data distribution $P_{\widetilde{\mathcal{X}}}$ and test-time data distribution $P_{\mathcal{X}}$ as:

$$\mathcal{H}^*_{aux} : h = \arg\min_{h \in \mathcal{H}} \epsilon_{P_{\widetilde{\mathcal{X}}}}(h, f), \mathcal{H}^*_{ood} : h = \arg\min_{h \in \mathcal{H}} \epsilon_{P_{\mathcal{X}}}(h, f). \tag{2}$$

It is worth noting that $\mathcal{H}^*_{ood} \subseteq \mathcal{H}^*_{aux}$ (given over-parameterized models)[2], reflecting the reality that hypotheses perform well on real-world OOD data also perform well on auxiliary outliers, given that auxiliary outliers are a subset of real-world OOD data. The generalization error of an OOD detector $h$ is defined as:

$$\text{GError}(h) = \epsilon_{x \sim \mathcal{P}_{\mathcal{X}}}(h, f). \tag{3}$$

Now, we present our first main result regarding OOD detection (training with auxiliary outliers).

**Theorem 1** *(Generalization Bound of OOD Detector).* *We let $\mathcal{D}_{train} = \mathcal{D}_{id} \cup \mathcal{D}_{aux}$, consisting of $N$ samples. For any hypothesis $h \in \mathcal{H}$ and $0 < \delta < 1$, with a probability of at least $1 - \delta$, the following inequality holds:*

$$GError(h) \leq \underbrace{\hat{\epsilon}_{x \sim \mathcal{P}_{\widetilde{\mathcal{X}}}}(h, f)}_{\text{empirical error}} + \underbrace{\epsilon(h, h^*_{aux})}_{\text{reducible error}} + \underbrace{\sup_{h \in \mathcal{H}^*_{aux}} \epsilon_{x \sim \mathcal{P}_{\mathcal{X}}}(h, h^*_{ood})}_{\text{outlier coverage error}} + \underbrace{\mathcal{R}_m(\mathcal{H})}_{\text{complexity}} + \sqrt{\frac{\ln(\frac{1}{\delta})}{2N}} + \beta, \tag{4}$$

*where $\hat{\epsilon}_{x \sim \mathcal{P}_{\widetilde{\mathcal{X}}}}(h, f)$ is the empirical error. We define $\epsilon(h, h^*_{aux}) = \int |\phi_{\mathcal{X}}(x) - \phi_{\widetilde{\mathcal{X}}}(x)||h(x) - h^*_{aux}(x)|dx$ as the reducible error, where $\phi_{\mathcal{X}}$ and $\phi_{\widetilde{\mathcal{X}}}$ is the density function of $\mathcal{P}_{\mathcal{X}}$ and $\mathcal{P}_{\widetilde{\mathcal{X}}}$ respectively. $\sup_{h \in \mathcal{H}^*_{aux}} \epsilon_{x \sim \mathcal{P}_{\mathcal{X}}}(h, h^*_{ood})$ is the* outlier coverage error, *$\mathcal{R}_m(h)$ represents the Rademacher complexity, and $\beta$ is the error related to ideal hypotheses. Notably, when $\beta$ is large, there exists no detector that performs well on $\mathcal{P}_{\mathcal{X}}$, making it unfeasible to find a good hypothesis through training with auxiliary outliers.*

Minimizing empirical risk optimizes the model $h$ to $h \in \mathcal{H}^*_{aux}$, leading to a reduction in the reducible error, which tends to zero. However, the inherent distribution shift error between auxiliary outliers and real-world OOD data remains constant. This limitation fundamentally restricts the generalization of OOD detection methods trained with auxiliary outliers. To address this limitation, we investigate the effect of outlier diversity.

## 3.3 Enhancing OOD Detection with Diverse Auxiliary Outliers

In this paper, the diversity refers to semantic diversity, where a formal definition is given as follows.

**Definition 1** *(Diversity of Outliers).* *We assume $\mathcal{X}_{aux}$ can be divided into distinct semantic subspaces: $\mathcal{X}_{aux} = \mathcal{X}^{y_1} \cup \mathcal{X}^{y_2} \cup \ldots \cup \mathcal{X}^{y_m}$, where each subspace $\mathcal{X}^{y_i}$ contains data points with label $y_i$. Given datasets $\mathcal{D}_{div}$ sampled from the distribution $\mathcal{P}_{\mathcal{X}_{div}}$, where $\mathcal{X}_{div} \subset \mathcal{X}_{ood}$ encompasses $\mathcal{X}_{aux}$ and a new subspace $\mathcal{X}_{new} = \mathcal{X}^{y_{m+1}} \ldots \cup \mathcal{X}^{y_n}$ with different semantic of $\mathcal{X}_{aux}$, i.e., $\mathcal{X}_{div} = \mathcal{X}_{aux} \cup \mathcal{X}_{new}$, we define that $\mathcal{D}_{div}$ is more diverse than $\mathcal{D}_{aux}$.*

Suppose we could use this diverse auxiliary outliers dataset for training, the ideal hypotheses achieved by training with $\mathcal{D}_{div}$ are denoted as

$$\mathcal{H}^*_{div} : h = \arg\min_{h \in \mathcal{H}} \epsilon_{x \sim \mathcal{P}_{\widetilde{\mathcal{X}}_{div}}}(h, f), \tag{5}$$

---

[1] OOD label function provides ground truth labels (OOD or ID) for the inputs.

[2] The over-parameterized setting enables the models to achieve a near-perfect fitting to a wide range of functions during training (the training loss is sufficiently small) (Zhang et al., 2021; Belkin et al., 2019).

with $\mathcal{P}_{\widetilde{\mathcal{X}}_{div}} = k_{train}\mathcal{P}_{\mathcal{X}_{id}} + (1 - k_{train})\mathcal{P}_{\mathcal{X}_{div}}$. Because $\mathcal{X}_{aux} \subset \mathcal{X}_{div}$ holds, the hypotheses performing well on $\mathcal{P}_{\mathcal{X}_{div}}$ also perform well on $\mathcal{P}_{\mathcal{X}_{aux}}$, giving rise to $\mathcal{H}^*_{div} \subset \mathcal{H}^*_{aux}$. Consequently, we have

$$\sup_{h \in \mathcal{H}^*_{div}} \epsilon_{x \sim \mathcal{P}_{\mathcal{X}}}(h, h^*_{ood}) \leq \sup_{h \in \mathcal{H}^*_{aux}} \epsilon_{x \sim \mathcal{P}_{\mathcal{X}}}(h, h^*_{ood}), \tag{6}$$

which indicates that training with a more diverse set of auxiliary outliers can reduce the outlier coverage error. Furthermore, effective training leads to sufficient small empirical error and reducible error, and the intrinsic complexity of the model remains constant. Consequently, a more diverse set of auxiliary outliers results in a lower generalization error bound. This theorem is formally presented as follows:

**Theorem 2** *(Diverse Outliers Enhance Generalization). Let $\mathcal{O}(GError(h))$ and $\mathcal{O}(GError(h_{div}))$ represent the upper bounds of the generalization error of detector training with vanilla auxiliary outliers $\mathcal{D}_{aux}$ and diverse auxiliary outliers $\mathcal{D}_{div}$, respectively. For any hypothesis $h$ and $h_{div}$ in $\mathcal{H}$, and $0 < \delta < 1$, with a probability of at least $1 - \delta$, the following inequality holds*

$$\mathcal{O}(GError(h_{div})) \leq \mathcal{O}(GError(h)). \tag{7}$$

**Remark.** Theorem 2 highlights that the diversity of the outlier set is a critical factor in reducing the upper bound of generalization error. However, despite the fundamental improvement in model generalization achieved by increasing the diversity of auxiliary outliers, collecting more auxiliary outliers is expensive, and the auxiliary outliers we can use are limited in practical scenarios, which hinders the application of outlier exposure methods for OOD detection. This raises an intuitive question: Can we enhance the diversity of a fixed set of auxiliary outliers to make better use of them?

## 4 METHOD: PROVABLE MIXUP OUTLIER

In this section, we show how mixup addresses the challenge of effective training when outlier diversity is limited. We begin with a theoretical analysis demonstrating the effectiveness of mixup in enhancing outlier diversity to improve OOD detection performance, providing a reliable guarantee for our method. Then, we introduce a simple yet effective framework implementing our mixup-based method to enhance OOD detection performance.

### 4.1 MIXUP FOR ENHANCING DIVERSITY OF AUXILIARY OUTLIERS

Mixup (Zhang et al., 2018) is a widely used machine learning technique to augment training data by creating synthetic samples. It involves generating virtual training examples (referred to as mixed samples) through linear interpolations between data points and their corresponding labels, given by:

$$\hat{x} = \lambda x_i + (1 - \lambda)x_j, \quad \hat{y} = \lambda y_i + (1 - \lambda)y_j, \tag{8}$$

where $(x_i, y_i)$ and $(x_j, y_j)$ are two samples drawn randomly from the empirical training distribution, and $\lambda \in [0, 1]$ is usually sampled from a Beta distribution denoted as $Beta(\alpha, \alpha)$. This technique assumes a linear relationship between semantics (labels) and features (in data), allowing us to create new samples that deviate from the semantics of the original ones by combining features from samples with distinct semantics. This assumption is formulated as follows:

**Assumption 1** *(Semantic Change under Mixup). Let $x_i$ and $x_j$ be any two data points from the input spaces $\mathcal{X}^{y_i}$ and $\mathcal{X}^{y_j}$, respectively, where $y_i$ and $y_j$ are their corresponding semantic labels and $y_i \neq y_j$. If $\zeta < \lambda < 1 - \zeta$, then there exists a positive value $\zeta$ such that the mixed data point $\hat{x} = \lambda x_i + (1 - \lambda)x_j$ does not belong to either set $\mathcal{X}^{y_i}$ or $\mathcal{X}^{y_j}$.*

This assumption suggests that we can enhance outlier diversity by generating new outliers with distinct semantics using mixup. Specifically, applying mixup to outliers in $\mathcal{X}_{aux}$ results in some mixed outliers having different semantics, suggesting that they belong to a novel subspace outside of $\mathcal{X}_{aux}$. Consequently, mixed outliers can be considered as samples from a broader subspace within the input space. As per Definition 1, mixed outliers exhibit greater diversity than the original outliers. This lemma is formally presented as follows:

**Lemma 1** *(Diversity Enhancement with Mixup). For a group of mixup transforms[3] $\mathcal{G}$ acting on the input space $\mathcal{X}_{aux}$ to generate an augmented input space $\mathcal{G}\mathcal{X}_{aux}$, defined as $\mathcal{G}\mathcal{X}_{aux} = \{\hat{x}|\hat{x} = \lambda x_1 + (1-\lambda)x_2; x_1, x_2 \in \mathcal{X}_{aux}, \lambda \in [0,1]\}$, the following relation holds:*

$$\mathcal{X}_{aux} \subset \mathcal{G}\mathcal{X}_{aux}. \tag{9}$$

Lemma 1 establishes that mixed outliers $\mathcal{D}_{mix}$ exhibits greater diversity compared to $\mathcal{D}_{aux}$, where $\mathcal{D}_{mix}$ is drawn from distribution $\mathcal{P}_{\mathcal{G}\mathcal{X}_{aux}}$. Consequently, according to Theorem 2, mixup outliers contribute to a reduction in generalization error. We can formalize this relationship as follows, and the detailed proofs can be found in *Appendix A*.

**Theorem 3** *(Mixed Outlier Enhances Generalization). Let $\mathcal{O}(GError(h))$ and $\mathcal{O}(GError(h_{mix}))$ represent the upper bounds of the generalization error of detector training with vanilla auxiliary outliers $\mathcal{D}_{aux}$ and mixed auxiliary outliers $\mathcal{D}_{mix}$, respectively. For any hypothesis $h$ and $h_{mix}$ in $\mathcal{H}$, and $0 < \delta < 1$, with a probability of at least $1 - \delta$, we have*

$$\mathcal{O}(GError(h_{mix})) \le \mathcal{O}(GError(h)). \tag{10}$$

Theorem 3 demonstrates that mixup enhances auxiliary outlier diversity, reducing the upper bound of generalization error in OOD detection, which provides a reliable guarantee of mixup's effectiveness in improving OOD detection. Next, we will provide an implementation of our method to show its practical effectiveness.

## 4.2 IMPLEMENTATION

We follow the basic settings of Chen et al. (2021). Considering a $(K + 1)$-way classifier network $F_\theta$, where the $(K + 1)$-th class label indicates OOD class, and $F_\theta(x)$ denotes the softmax output of $F_\theta$ for input $x$. We denote the OOD score as $c(x) = F_\theta(x)_{K+1}$. At test time, we construct the OOD detector $G(x)$ using:

$$G(x) = \begin{cases} \text{OOD}, & \text{if } c(x) \ge \gamma \\ \text{ID}, & \text{if } c(x) < \gamma \end{cases} \tag{11}$$

where $\gamma$ is the threshold. Given an input detected as ID by $G(x)$, its ID label can be obtained using $\hat{F}(x)$:

$$\hat{F}(x) = \underset{y \in \{1,2,...,K\}}{\arg\max} F(x)_y. \tag{12}$$

The training objective is given by:

$$\arg\min_\theta \mathbb{E}_{(x,y)\sim\mathcal{D}_{id}}[l(F_\theta(x), y)] + \omega \cdot \mathbb{E}_{x'\sim\mathcal{D}_{mix}}[l(F_\theta(x'), K+1)], \tag{13}$$

where $l$ is the cross entropy loss and $\omega$ is a hyperparameter which controls the strength of regularization. Constructing the mixed outliers dataset $\mathcal{D}_{mix}$ is computationally expensive due to the need to cover all possible data point combinations and $\lambda$ values for mixup. To address this, we use an online approach. In each training epoch, we randomly sample $N$ data points from the large pool of auxiliary dataset to create $\mathcal{S}$. Mixup is then applied to part of $\mathcal{S}$ to generate $\mathcal{S}_{mix}$, which can be regarded as drawn from $\mathcal{P}_{\mathcal{G}\mathcal{X}_{aux}}$ when considering the overall training process.

Considering that not all mixed outliers are equally valuable for model training, as some are easily detectable, it is important to identify and prioritize those outliers that provide more valuable information. We focus on mining informative mixed outliers located at the classification boundary between ID and OOD data for training. We evaluate the set $\mathcal{S}_{mix}$ using the existing model and select outliers with the lowest OOD score which are the most informative. These selected mixed outliers constitute the mixed outliers dataset $\mathcal{D}_{mix}$ that is used for the training objective (13). The whole pseudocode of the proposed method is shown in Alg. 1.

**Remark.** Mixup has a unique advantage in diversifying auxiliary outliers as it can generate outliers with distinct semantics from the originals without much computational overhead, while conventional data augmentation methods often produce outliers closely aligned with the original data, as they simply perturb existing points without introducing semantic variation.

---

[3] The set of all possible combinations of data points and all possible values of $\lambda$ for mixup.

---

**Algorithm 1** Provable Mixup Outlier

---

**Input:** ID dataset $\mathcal{D}_{id}$, outliers dataset $\mathcal{D}_{aux}$, pool size $N$, informative fraction $\mu$, hyperparameter of mixup ratio $\sigma$, hyperparameter of Beta distribution $\alpha$;
**Output:** ID classifier $\hat{F}$, OOD detector $G$;
**for** *each iteration* **do**
      // Randomly sample $N$ outliers from $\mathcal{D}_{aux}$ to create subset $\mathcal{S}$.
      $\mathcal{S} \leftarrow \{s_i \,|\, s_i \in \mathcal{D}_{aux}, \; i = 1, 2, ..., N\}$;
      // Copy $\mathcal{S}$ and random shuffle to create $\mathcal{S}'$.
      $\mathcal{S}' \leftarrow \text{shuffle}(\mathcal{S})$;
      // Apply mixup on $\mathcal{S}$ and $\mathcal{S}'$ to obtain mixed candidate set $\mathcal{S}_{mix}$.
      Sample $p \sim \text{Uniform}(0, 1)$;
      $\mathcal{S}_{mix} \leftarrow \{\hat{x}_i \,|\, \hat{x}_i = \lambda x_i + (1 - \lambda)x_i'; \; x \in \mathcal{S}, \; x' \in \mathcal{S}'; \; i = 1, 2, ..., N;$
      **if** $p \leq \sigma$, **then** $\lambda \sim \text{Beta}(\alpha, \alpha)$, **else** $\lambda = 1\}$ ;
      // Compute OOD scores on $\mathcal{S}_{mix}$ using the current model.
      $\mathcal{C} \leftarrow \{c \,|\, c = F_\theta(x)_{k+1}, \; x \in \mathcal{S}_{mix}\}$;
      // Sort $\mathcal{S}_{mix}$ in order of OOD scores $\mathcal{C}$ from least to greatest.
      $\mathcal{S}_{mix} \leftarrow \mathcal{S}_{mix}[\text{argsort}(\mathcal{C})]$;
      // Select the first $\mu N$ samples from $\mathcal{S}$ as the mixed auxiliary outliers $\mathcal{D}_{mix}$.
      $\mathcal{D}_{mix} \leftarrow \mathcal{S}_{mix}[: \mu N]$;
      Train $F_\theta$ for one epoch using the training objective of Eq. 13;
Build $\hat{F}$ and $G$ according to Eq. 12 and Eq. 11;

---

## 5 EXPERIMENTS

In this section, we outline our experimental setup and conduct experiments on common OOD detection benchmarks to answer the following questions: **Q1.** Effectiveness: Does our method outperform its counterparts? **Q2.** Reliability: Is our theory truly dependable? **Q3.** Ablation study (I): What is the key factor contributing to performance improvement in our method? **Q4.** Ablation study (II): Does ProMix truly offer a distinct advantage over other data augmentation methods?

### 5.1 EXPERIMENTAL SETUP

We briefly present the experimental setup here, including the experimental datasets, training details and evaluation metrics. Further experimental setup details can be found in *Appendix B*. It is worth noting that we are committed to open-sourcing the code related to our research after publication.

**Datasets.** We use common benchmarks from previous work (Chen et al., 2021; Ming et al., 2022), including CIFAR-10 and CIFAR-100 (Krizhevsky et al., 2009). A downsampled version of ImageNet (ImageNet-RC) (Deng et al., 2009) is used as auxiliary outliers. For OOD test sets, we employ diverse image datasets: SVHN (Netzer et al., 2011), Textures (Cimpoi et al., 2014), Places365 (Zhou et al., 2016), LSUN-crop, LSUN-resize (Yu et al., 2015), and iSUN (Xu et al., 2015).

**Training details.** We use DenseNet-101 (Huang et al., 2017) as the backbone for all methods, employing stochastic gradient descent with Nesterov momentum (momentum = 0.9) over 100 epochs. The initial learning rate of 0.1 decreases by a factor of 0.1 at 50, 75, and 90 epochs. Batch sizes are 64 for ID data and 128 for OOD data. For mixup, we set $\alpha = 1$, $\sigma = 0.5$. In the greedy sampling phase, We assign the values $N = 400,000$ and $\mu N = 100,000$. Experiments are run over five times to report the means and standard deviations.

**Evaluation metrics.** Following common practice, we report: (1) OOD false positive rate (FPR95) at 95% true positive rate for ID samples (Liang et al., 2018a), (2) the area under the receiver operating characteristic curve (AUROC) (Davis & Goadrich, 2006), (3) the area under the precision-recall curve (AUPR) (Manning & Schutze, 1999). We also provide ID classification accuracy (ID-ACC).

### 5.2 EXPERIMENTAL RESULTS AND DISCUSSION

**Q1 Effective.** Our method outperforms existing competitive methods, establishing *state-of-the-art* performance both on CIFAR-10 and CIFAR-100 datasets. Table 1 provides a comprehensive

Table 1: **Main results.** Comparison with competitive OOD detection methods trained with the same DenseNet backbone. The performance is averaged (%) over six OOD test datasets from Section 5.1. Some baseline results are sourced from Ming et al. (2022). The best results are in red, with standard deviations are provided in the *Appendix C.1*.

| $D_{in}$ | Method | **FPR95** ($\downarrow$) | **AUROC** ($\uparrow$) | **AUPR** ($\uparrow$) | ID-ACC | w./w.o. $\mathcal{D}_{aux}$ | informative | diverse |
|---|---|---|---|---|---|---|---|---|
| | MSP | 58.98 | 90.63 | 93.18 | 94.39 | $\times$ | NA | NA |
| | ODIN | 26.55 | 94.25 | 95.34 | 94.39 | $\times$ | NA | NA |
| | Mahalanobis | 29.47 | 89.96 | 89.70 | 94.39 | $\times$ | NA | NA |
| | Energy | 28.53 | 94.39 | 95.56 | 94.39 | $\times$ | NA | NA |
| | SSD+ | 7.22 | 98.48 | 98.59 | NA | $\times$ | NA | NA |
| CIFAR-10 | OE | 9.66 | 98.34 | 98.55 | 94.12 | $\checkmark$ | $\times$ | $\times$ |
| | SOFL | 5.41 | 98.98 | 99.10 | 93.68 | $\checkmark$ | $\times$ | $\times$ |
| | CCU | 8.78 | 98.41 | 98.69 | 93.97 | $\checkmark$ | $\times$ | $\times$ |
| | Energy (w. $D_{aux}$) | 4.62 | 98.93 | 99.12 | 92.92 | $\checkmark$ | $\times$ | $\times$ |
| | NTOM | 4.00 | 99.09 | 98.61 | 94.26 | $\checkmark$ | $\checkmark$ | $\times$ |
| | POEM | 2.54 | 99.40 | **99.50** | 93.49 | $\checkmark$ | $\checkmark$ | $\times$ |
| | **ProMix (ours)** | **2.18** | **99.43** | 99.01 | 94.32 | $\checkmark$ | $\checkmark$ | $\checkmark$ |
| | MSP | 80.30 | 73.13 | 76.97 | 74.05 | $\times$ | NA | NA |
| | ODIN | 56.31 | 84.89 | 85.88 | 74.05 | $\times$ | NA | NA |
| | Mahalanobis | 47.89 | 85.71 | 87.15 | 74.05 | $\times$ | NA | NA |
| | Energy | 65.87 | 81.50 | 84.07 | 74.05 | $\times$ | NA | NA |
| | SSD+ | 38.32 | 88.91 | 89.77 | NA | $\times$ | NA | NA |
| CIFAR-100 | OE | 19.54 | 94.93 | 95.26 | 74.25 | $\checkmark$ | $\times$ | $\times$ |
| | SOFL | 19.32 | 96.32 | 96.99 | 73.93 | $\checkmark$ | $\times$ | $\times$ |
| | CCU | 19.27 | 95.02 | 95.41 | 74.49 | $\checkmark$ | $\times$ | $\times$ |
| | Energy (w. $D_{aux}$) | 19.25 | 96.68 | 97.44 | 72.39 | $\checkmark$ | $\times$ | $\times$ |
| | NTOM | 18.77 | 96.69 | 96.49 | 74.52 | $\checkmark$ | $\checkmark$ | $\times$ |
| | POEM | 15.14 | 97.79 | 98.31 | 73.41 | $\checkmark$ | $\checkmark$ | $\times$ |
| | **ProMix (ours)** | **10.37** | **98.03** | **98.63** | 74.26 | $\checkmark$ | $\checkmark$ | $\checkmark$ |

comparison with various methods grouped as follows: (1) ID-only training: MSP (Hendrycks & Gimpel, 2017), ODIN (Liang et al., 2018b), Mahalanobis (Lee et al., 2018b), Energy (Wang et al., 2021a); (2) Utilizing auxiliary outliers (without outlier mining): OE (Hendrycks & andThomas G. Dietterich, 2019), SOFL (Mohseni et al., 2020), CCU (Meinke & Hein, 2019), Energy with outlier (Wang et al., 2021a); (3) Outlier mining: NTOM (Chen et al., 2021), POEM (Ming et al., 2022). Specifically, when compared to the best baseline, ProMix achieves a reduction in terms of FPR95 of $0.36\%$ and $4.77\%$ on CIFAR-10 and CIFAR-100, respectively, which means a relative error reductions of $14.2\%$ and $31.5\%$. Furthermore, in comparison to NTOM (Chen et al., 2021), which employs the same sampling strategy as our method, we achieve FPR95 reductions of $1.81\%$ and $8.4\%$ on CIFAR-10 and CIFAR-100 respectively. These reductions correspond to relative error reductions of $45.3\%$ and $44.8\%$. These notable improvements can be attributed to the enhanced diversity in auxiliary outliers offered by ProMix, which serves to reduce the generalization error bound and significantly enhances OOD detection performance.

**Q2 Reliability.** Our theory demonstrates that augmenting the semantic diversity of auxiliary outliers leads to a substantial enhancement in performance. To confirm this, we manipulate the number of classes within the outlier dataset while keeping the data size constant. This manipulation allows us to regulate the level of semantic diversity. Figure 2(a) illustrates noticeable performance improvements as semantic diversity increases. Moreover, to emphasize that performance is influenced primarily by semantic diversity rather than data size (sample diversity), we maintain a consistent number of classes within the outlier dataset but decrease the data size. Figure 2(b) demonstrates that data size exerts limited influence on performance, thereby establishing the reliability of our theoretical framework. Furthermore, it is worth noting that our proposed method consistently achieves significant performance improvements across a range of diversity and data size settings, which further validates the reliability and effectiveness of our approach.

**Q3 Ablation Study (I).** We conduct an ablation study to explore the impact of various components within our method. Our approach primarily relies on mixup outliers and a greedy sampling strategy. Table 5 highlights two key findings: (1) Mixup outliers play a substantial role in enhancing OOD detection performance by introducing diversity among outliers. (2) Additionally, the integration of the greedy sampling strategy results in a performance improvement, which is likely attributed to its ability to enhance the efficiency of outlier utilization.

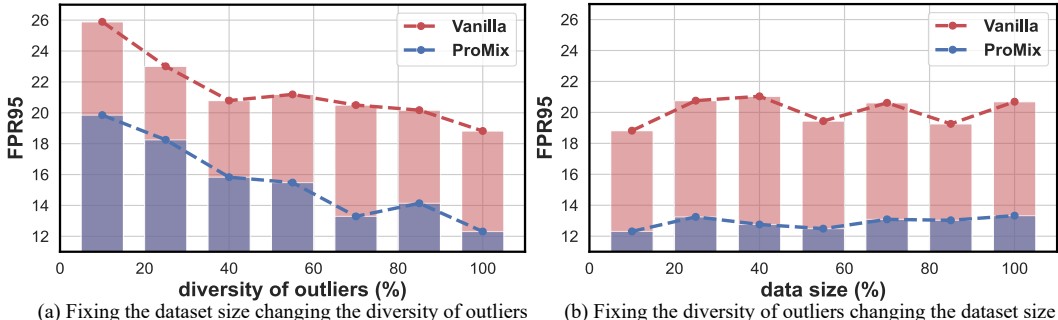

(a) Fixing the dataset size changing the diversity of outliers  (b) Fixing the diversity of outliers changing the dataset size

Figure 2: Comparing the performance of OOD detection methods on CIFAR-100 with different types of auxiliary outliers. (a) Varying outlier diversity, with the x-axis representing the proportion in the original outlier dataset classes. (b) Varying outlier dataset sizes, with the x-axis reflecting the proportion of the original outlier dataset's data size. See *Appendix B.2* for more details.

Table 2: **Ablation study on module contributions.** Performance averaged (%) over six OOD test datasets from Section 5.1. The best results are in **red**, with standard deviations in the *Appendix C.1*.

| Mixup | Greedy sample | CIFAR-10 | | | | CIFAR-100 | | | |
|---|---|---|---|---|---|---|---|---|---|
| | | FPR ($\downarrow$) | AUROC ($\uparrow$) | AUPR ($\uparrow$) | ID-ACC | FPR ($\downarrow$) | AUROC ($\uparrow$) | AUPR ($\uparrow$) | ID-ACC |
| $\times$ | $\times$ | 5.14 | 98.91 | 98.89 | 94.40 | 20.69 | 95.90 | 95.65 | 74.27 |
| $\times$ | $\checkmark$ | 4.00 | 99.09 | 98.61 | 94.26 | 18.77 | 96.69 | 96.49 | 74.52 |
| $\checkmark$ | $\times$ | 3.28 | 99.23 | **99.45** | 94.30 | 13.33 | 97.35 | **98.66** | 74.28 |
| $\checkmark$ | $\checkmark$ | **2.18** | **99.43** | 99.01 | 94.32 | **10.37** | **98.03** | 98.63 | 74.26 |

Table 3: **Ablation on data augmentation method.** Performance averaged (%) over six OOD test datasets from Section 5.1. The best results are in **red**, with standard deviations in the *Appendix C.1*.

| | Semantic change ? | CIFAR-10 | | | | CIFAR-100 | | | |
|---|---|---|---|---|---|---|---|---|---|
| | | FPR ($\downarrow$) | AUROC ($\uparrow$) | AUPR ($\uparrow$) | ID-ACC | FPR ($\downarrow$) | AUROC ($\uparrow$) | AUPR ($\uparrow$) | ID-ACC |
| Noise | $\times$ | 7.84 | 98.56 | 98.87 | 94.02 | 26.01 | 94.87 | 96.06 | 74.69 |
| Cutout | $\times$ | 8.83 | 98.45 | 98.37 | 94.27 | 24.36 | 95.01 | 94.91 | 74.55 |
| Vanilla | NA | 5.41 | 98.91 | 98.89 | 94.40 | 20.69 | 95.90 | 95.65 | 74.27 |
| Cutmix | $\checkmark$ | 5.76 | 98.95 | 99.08 | 94.37 | 21.23 | 96.03 | 96.34 | 74.45 |
| ProMix (ours) | $\checkmark$ | **3.28** | **99.23** | **99.45** | 94.30 | **13.33** | **97.35** | **98.66** | 74.28 |

**Q4 Ablation study (II).** Finally, we conduct the ablation study to compare ProMix with other data augmentation methods. Table 3 shows that adding noise or using cutout, both of which do not change the semantics of outliers, do not improve performance. Instead, they have the potential to introduce extraneous noise, thereby degrading the original data, and leading to a deterioration in performance. For mixup methods, cutmix introduces noise at clipping boundaries, which may result in overfitting and subsequently inferior performance compared to ProMix. For more details of the comparative methods, please refer to *Appendix B.3*.

## 6 CONCLUSIONS AND FUTURE WORK

In this study, we demonstrate that the performance of OOD detection methods is hindered by the distribution shift between unknown test OOD data and auxiliary outliers. Through rigorous theoretical analysis, we demonstrate that enhancing the diversity of auxiliary outliers can effectively mitigate this problem. However, constrained by limited access to auxiliary outliers and the high cost of data collection, we introduce ProMix, an effective method that enhances the diversity of auxiliary outliers and significantly improves model performance. The effectiveness of ProMix is supported by both theoretical analysis and empirical evidence. Further exploration could explore different sampling strategies for ProMix. Additionally, our theoretical analysis does not investigate the relationship between informative outliers and enhanced model performance. Future work will focus on developing a comprehensive theory to elaborate the role of diversity and informative outliers on OOD detection performance.

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

# Appendix

## CONTENTS

## A  THEORETICAL ANALYSIS

In this section, we provide detailed proofs of our theories and the proposed method, including the establishment of the generalization error bound for OOD detection (Theorem 1), a more diverse set of auxiliary outliers leads to a reduced generalization error (Theorem 2), and the proof of diversity enhancement with mixup (Lemma 1).

### A.1  PROOF OF THEOREM 1

In this section, we analyze the generalization error of the OOD detector training with auxiliary outliers. First, we recall the setting from Sec. 3.1, our goal is to train a detector with auxiliary outliers that can perform well on real-world OOD data. In other words, we aim to train a model on data sampled from $\mathcal{P}_{\widetilde{\mathcal{X}}} = k_{train}\mathcal{P}_{\mathcal{X}_{id}} + (1 - k_{train})\mathcal{P}_{\mathcal{X}_{aux}}$ to obtain a reliable hypothesis $h$ that can effectively generalize to the unknown test-time distribution $\mathcal{P}_{\mathcal{X}} = k_{test}\mathcal{P}_{\mathcal{X}_{id}} + (1 - k_{test})\mathcal{P}_{\mathcal{X}_{ood}}$.

Next, we develop bounds on the OOD detection performance of a detector training with auxiliary outliers, which can be formulated as follow:

*(**Generalization Bound of OOD Detector**). Let $\mathcal{D}_{train} = \mathcal{D}_{id} \cup \mathcal{D}_{aux}$, consisting of $N$ samples. For any hypothesis $h \in \mathcal{H}$ and $0 < \delta < 1$, with a probability of at least $1 - \delta$, the following inequality holds:*

$$GError(h) \leq \underbrace{\hat{\epsilon}_{x \sim \mathcal{P}_{\widetilde{\mathcal{X}}}}(h, f)}_{empirical\ error} + \underbrace{\epsilon(h, h^*_{aux})}_{reducible\ error} + \underbrace{\sup_{h \in \mathcal{H}^*_{aux}} \epsilon_{x \sim \mathcal{P}_{\mathcal{X}}}(h, h^*_{ood})}_{outlier\ coverage\ error} + \underbrace{\mathcal{R}_m(\mathcal{H})}_{complexity} + \sqrt{\frac{\ln(\frac{1}{\delta})}{2N}} + \beta, \quad (14)$$

where $\hat{\epsilon}_{x\sim\mathcal{P}_{\widetilde{\mathcal{X}}}}(h,f)$ is the empirical error. We define $\epsilon(h,h^*_{aux}) = \int |\phi_{\mathcal{X}}(x) - \phi_{\widetilde{\mathcal{X}}}(x)||h(x) - h^*_{aux}(x)|dx$ is the reducible error, $\phi_{\mathcal{X}}$ and $\phi_{\widetilde{\mathcal{X}}}$ is the density function of $\mathcal{P}_{\mathcal{X}}$ and $\mathcal{P}_{\widetilde{\mathcal{X}}}$ respectively. $\sup_{h\in\mathcal{H}^*_{aux}} \epsilon_{x\sim\mathcal{P}_{\mathcal{X}}}(h,h^*_{ood})$ is the outlier coverage color, $\mathcal{R}_m(h)$ represents the Rademacher complexity, $\beta$ is the error related to ideal hypotheses. The roadmap of our analysis is as follows:

**Roadmap.** We first show how to bound the OOD detection error in terms of the generalization error on $\mathcal{P}_{\widetilde{\mathcal{X}}}$ and the maximum distribution shift error as well as the reducible error which can be reduced to a small value as the model is optimized. Then, we study the generalization bound from the perspective of Rademacher complexity. We use complexity-based learning theory to quantify the generalization error on $\mathcal{P}_{\widetilde{\mathcal{X}}}$. In the end, we bound the OOD detection generalization error in terms of the empirical error on the training data, the reducible error, the maximum distribution shift error, and the complexity. We also provide detailed proof steps as follows:

**Proof.** This proof relies on the triangle inequality for classification error (Ben-David et al., 2006; Crammer et al., 2006), which implies that for any labeling functions $f_1$, $f_2$, and $f_3$, we have $\epsilon(f_1, f_2) \le \epsilon(f_1, f_3) + \epsilon(f_2, f_3)$.

$$GError(h) = \epsilon_{x\sim\mathcal{P}_{\mathcal{X}}}(h,f)$$
$$\le \epsilon_{x\sim\mathcal{P}_{\mathcal{X}}}(h,h^*_{ood}) + \epsilon_{x\sim\mathcal{P}_{\mathcal{X}}}(h^*_{ood},f)$$
$$= \epsilon_{x\sim\mathcal{P}_{\mathcal{X}}}(h,h^*_{ood}) + \epsilon_{x\sim\mathcal{P}_{\mathcal{X}}}(h^*_{ood},f) + \epsilon_{x\sim\mathcal{P}_{\widetilde{\mathcal{X}}}}(h,h^*_{ood}) - \epsilon_{x\sim\mathcal{P}_{\widetilde{\mathcal{X}}}}(h,h^*_{ood})$$
$$= \epsilon_{x\sim\mathcal{P}_{\widetilde{\mathcal{X}}}}(h,h^*_{ood}) + \epsilon_{x\sim\mathcal{P}_{\mathcal{X}}}(h^*_{ood},f) + \epsilon_{x\sim\mathcal{P}_{\mathcal{X}}}(h,h^*_{ood}) - \epsilon_{x\sim\mathcal{P}_{\widetilde{\mathcal{X}}}}(h,h^*_{ood})$$
$$\le \epsilon_{x\sim\mathcal{P}_{\widetilde{\mathcal{X}}}}(h,f) + \epsilon_{x\sim\mathcal{P}_{\widetilde{\mathcal{X}}}}(h^*_{ood},f) + \epsilon_{x\sim\mathcal{P}_{\mathcal{X}}}(h^*_{ood},f) + \epsilon_{x\sim\mathcal{P}_{\mathcal{X}}}(h,h^*_{ood}) - \epsilon_{x\sim\mathcal{P}_{\widetilde{\mathcal{X}}}}(h,h^*_{ood})$$

Let $\phi_{\mathcal{X}}$ and $\phi_{\widetilde{\mathcal{X}}}$ be the density functions of $\mathcal{P}_{\mathcal{X}}$ and $\mathcal{P}_{\widetilde{\mathcal{X}}}$, respectively.

$$GError(h) \le \epsilon_{x\sim\mathcal{P}_{\widetilde{\mathcal{X}}}}(h,f) + \epsilon_{x\sim\mathcal{P}_{\widetilde{\mathcal{X}}}}(h^*_{ood},f) + \epsilon_{x\sim\mathcal{P}_{\mathcal{X}}}(h^*_{ood},f)$$
$$+ \int \phi_{\mathcal{X}}(x)|h(x) - h^*_{ood}(x)|\, dx - \int \phi_{\widetilde{\mathcal{X}}}(x)|h(x) - h^*_{ood}(x)|\, dx$$

$$\le \epsilon_{x\sim\mathcal{P}_{\widetilde{\mathcal{X}}}}(h,f) + \epsilon_{x\sim\mathcal{P}_{\widetilde{\mathcal{X}}}}(h^*_{ood},f) + \epsilon_{x\sim\mathcal{P}_{\mathcal{X}}}(h^*_{ood},f) + \int |\phi_{\mathcal{X}}(x) - \phi_{\widetilde{\mathcal{X}}}(x)|\,|h(x) - h^*_{ood}(x)|\, dx$$

$$\le \epsilon_{x\sim\mathcal{P}_{\widetilde{\mathcal{X}}}}(h,f) + \epsilon_{x\sim\mathcal{P}_{\widetilde{\mathcal{X}}}}(h^*_{ood},f) + \epsilon_{x\sim\mathcal{P}_{\mathcal{X}}}(h^*_{ood},f) + \int |\phi_{\mathcal{X}}(x) - \phi_{\widetilde{\mathcal{X}}}(x)|\,|h(x) - h^*_{aux}(x)|\, dx$$
$$+ \int |\phi_{\mathcal{X}}(x) - \phi_{\widetilde{\mathcal{X}}}(x)|\,|h^*_{aux}(x) - h^*_{ood}(x)|\, dx$$

$$\le \epsilon_{x\sim\mathcal{P}_{\widetilde{\mathcal{X}}}}(h,f) + \epsilon_{x\sim\mathcal{P}_{\widetilde{\mathcal{X}}}}(h^*_{ood},f) + \epsilon_{x\sim\mathcal{P}_{\mathcal{X}}}(h^*_{ood},f) + \int |\phi_{\mathcal{X}}(x) - \phi_{\widetilde{\mathcal{X}}}(x)|\,|h(x) - h^*_{aux}(x)|\, dx$$
$$+ \int \phi_{\mathcal{X}}(x)\,|h^*_{aux}(x) - h^*_{ood}(x)|\, dx + \int \phi_{\widetilde{\mathcal{X}}}(x)\,|h^*_{aux}(x) - h^*_{ood}(x)|\, dx$$

$$= \epsilon_{x\sim\mathcal{P}_{\widetilde{\mathcal{X}}}}(h,f) + \epsilon_{x\sim\mathcal{P}_{\widetilde{\mathcal{X}}}}(h^*_{ood},f) + \epsilon_{x\sim\mathcal{P}_{\mathcal{X}}}(h^*_{ood},f) + \int |\phi_{\mathcal{X}}(x) - \phi_{\widetilde{\mathcal{X}}}(x)|\,|h(x) - h^*_{aux}(x)|\, dx$$
$$+ \epsilon_{x\sim\mathcal{P}_{\mathcal{X}}}(h^*_{aux},h^*_{ood}) + \epsilon_{x\sim\mathcal{P}_{\widetilde{\mathcal{X}}}}(h^*_{aux},h^*_{ood})$$

$$\le \epsilon_{x\sim\mathcal{P}_{\widetilde{\mathcal{X}}}}(h,f) + \epsilon_{x\sim\mathcal{P}_{\widetilde{\mathcal{X}}}}(h^*_{ood},f) + \epsilon_{x\sim\mathcal{P}_{\mathcal{X}}}(h^*_{ood},f) + \int |\phi_{\mathcal{X}}(x) - \phi_{\widetilde{\mathcal{X}}}(x)|\,|h(x) - h^*_{aux}(x)|\, dx$$
$$+ \epsilon_{x\sim\mathcal{P}_{\mathcal{X}}}(h^*_{aux},h^*_{ood}) + \epsilon_{x\sim\mathcal{P}_{\widetilde{\mathcal{X}}}}(h^*_{aux},f) + \epsilon_{x\sim\mathcal{P}_{\widetilde{\mathcal{X}}}}(h^*_{ood},f)$$

We denote $\beta_1 = \min_{h\in\mathcal{H}}\epsilon_{x\sim\mathcal{P}_{\mathcal{X}}}(h,f)$, $\beta_2 = \min_{h\in\mathcal{H}}\epsilon_{x\sim\mathcal{P}_{\widetilde{\mathcal{X}}}}(h,f)$ as the error of $h^*_{ood}$ and $h^*_{aux}$ on $\mathcal{P}_{\mathcal{X}}$ and $\widetilde{\mathcal{P}}_{\mathcal{X}}$,

$$GError(h) \le \epsilon_{x\sim\mathcal{P}_{\widetilde{\mathcal{X}}}}(h,f) + \beta_1 + \beta_2 + \int |\phi_{\mathcal{X}}(x) - \phi_{\widetilde{\mathcal{X}}}(x)|\,|h(x) - h^*_{aux}(x)|\, dx$$
$$+ \epsilon_{x\sim\mathcal{P}_{\mathcal{X}}}(h^*_{aux},h^*_{ood}) + \beta_2 + \beta_2$$

We denote $1/4 * \beta = max\{\beta_1, \beta_2\}$, so

$$GError(h) \leq \epsilon_{x \sim \mathcal{P}_{\widetilde{\mathcal{X}}}}(h, f) + \int |\phi_{\mathcal{X}}(x) - \phi_{\widetilde{\mathcal{X}}}(x)||h(x) - h^*_{aux}(x)| \, dx + \epsilon_{x \sim \mathcal{P}_{\mathcal{X}}}(h^*_{aux}, h^*_{ood}) + \beta$$

Consider an upper bound on the distribution shift error $\epsilon_{x \sim \mathcal{P}_{\mathcal{X}}}(h^*_{aux}, h^*_{ood})$

$$GError(h) \leq \epsilon_{x \sim \mathcal{P}_{\widetilde{\mathcal{X}}}}(h, f) + \int |\phi_{\mathcal{X}}(x) - \phi_{\widetilde{\mathcal{X}}}(x)||h(x) - h^*_{aux}(x)| \, dx$$
$$+ \sup_{h \in \mathcal{H}^*_{aux}} \epsilon_{x \sim \mathcal{P}_{\mathcal{X}}}(h, h^*_{ood}) + \beta,$$

Next, we recap the Rademacher complexity measure for model complexity. We use complexity-based learning theory (Bartlett & Mendelson, 2002) (Theorem 8) to quantify the generalization error. Let $\mathcal{D}_{train} = \mathcal{D}_{id} \cup \mathcal{D}_{aux}$ consisting of $N$ samples, $\hat{\epsilon}_{x \sim \mathcal{P}_{\widetilde{\mathcal{X}}}}(h, f)$ is the empirical error of $h$. Then for any hypothesis $h$ in $\mathcal{H}$ (i.e., $\mathcal{H} : \mathcal{X} \to \{0, 1\}, h \in \mathcal{H}$) and $1 > \delta > 0$, with probability at least $1 - \delta$, we have

$$\epsilon_{x \sim \mathcal{P}_{\widetilde{\mathcal{X}}}}(h, f) \leq \hat{\epsilon}_{x \sim \mathcal{P}_{\widetilde{\mathcal{X}}}}(h, f) + \mathcal{R}_m(\mathcal{H}) + \sqrt{\frac{ln(\frac{1}{\delta})}{2N}}$$

where $\mathcal{R}_m(\mathcal{H})$ is the Rademacher complexities. Finally, it holds with a probability of at least $1 - \delta$ that

$$\epsilon_{x \sim \mathcal{P}_{\mathcal{X}}}(h, f) \leq \underbrace{\hat{\epsilon}_{x \sim \mathcal{P}_{\widetilde{\mathcal{X}}}}(h, f)}_{\text{empirical error}} + \underbrace{\epsilon(h, h^*_{aux})}_{\text{reducible error}} + \underbrace{\sup_{h \in \mathcal{H}^*_{aux}} \epsilon_{x \sim \mathcal{P}_{\mathcal{X}}}(h, h^*_{ood})}_{\text{outlier coverage error}} + \underbrace{\mathcal{R}_m(\mathcal{H})}_{\text{complexity}} + \sqrt{\frac{ln(\frac{1}{\delta})}{2N}} + \beta$$

where $\epsilon(h, h^*_{aux}) = \int |\phi_{\mathcal{X}}(x) - \phi_{\widetilde{\mathcal{X}}}(x)| \, |h(x) - h^*_{aux}(x)| \, dx$ represents the reducible error and $\beta$ is the error related to ideal hypotheses. When $\beta$ is large, then there is no detector that performs well. Therefore, we cannot hope to find a good hypothesis by training with auxiliary outliers.

## A.2 PROOF OF THEOREM 2

In this section, we proof that diverse outliers enhance generalization, which can be formulated as follows:

*Let $\mathcal{O}(GError(h))$ and $\mathcal{O}(GError(h_{div}))$ represent the upper bounds of the generalization error of detector training with vanilla auxiliary outliers $\mathcal{D}_{aux}$ and diverse auxiliary outliers $\mathcal{D}_{div}$, respectively. For any hypothesis $h$ and $h_{div}$ in $\mathcal{H}$, and $0 < \delta < 1$, with a probability of at least $1 - \delta$, the following inequality holds*

$$\mathcal{O}(GError(h_{div})) \leq \mathcal{O}(GError(h)). \tag{15}$$

The detailed proof proceeds as follows:

**Proof.** At first, we prove that diverse outliers correspond to a smaller outlier coverage error than vanilla outliers. Because $\mathcal{X}_{aux} \subset \mathcal{X}_{div}$ holds, the hypotheses performing well on $\mathcal{P}_{\mathcal{X}_{div}}$ also perform well on $\mathcal{P}_{\mathcal{X}_{aux}}$, giving rise to $\mathcal{H}^*_{div} \subset \mathcal{H}^*_{aux}$.

$$\sup_{h \in \mathcal{H}^*_{div}} \epsilon_{x \sim \mathcal{P}_{\mathcal{X}}}(h, h^*_{ood}) \leq \max\{ \sup_{h \in \mathcal{H}^*_{div}} \epsilon_{x \sim \mathcal{P}_{\mathcal{X}}}(h, h^*_{ood}), \sup_{h \in \mathcal{H}^*_{aux} - \mathcal{H}^*_{div}} \epsilon_{x \sim \mathcal{P}_{\mathcal{X}}}(h, h^*_{ood})\},$$

note that

$$\max\{ \sup_{h \in \mathcal{H}^*_{div}} \epsilon_{x \sim \mathcal{P}_{\mathcal{X}}}(h, h^*_{ood}), \sup_{h \in \mathcal{H}^*_{aux} - \mathcal{H}^*_{div}} \epsilon_{x \sim \mathcal{P}_{\mathcal{X}}}(h, h^*_{ood})\} = \sup_{h \in \mathcal{H}^*_{aux}} \epsilon_{x \sim \mathcal{P}_{\mathcal{X}}}(h, h^*_{ood}),$$

Consequently, we have

$$\sup_{h \in \mathcal{H}^*_{div}} \epsilon_{x \sim \mathcal{P}_{\mathcal{X}}}(h, h^*_{ood}) \leq \sup_{h \in \mathcal{H}^*_{aux}} \epsilon_{x \sim \mathcal{P}_{\mathcal{X}}}(h, h^*_{ood}). \tag{16}$$

Furthermore, model effective training leads to small empirical error and small reducible error, if we continue to use the same model architecture, the intrinsic complexity of the model $\mathcal{R}_m(\mathcal{H})$ remains invariant, consider that $\beta$ is a small constant value, therefore, it holds that

$$\mathcal{O}(GError(h_{div})) \leq \mathcal{O}(GError(h)), \tag{17}$$

with a probability of at least $1 - \delta$.

A.3    PROOF OF LEMMA 1

In this section, we give the proof of the Lemma 1, which can be formalized as follow:

**(Diversity Enhancement with Mixup).** *For a group of mixup transforms[4] $\mathcal{G}$ acting on the input space $\mathcal{X}_{aux}$ to generate an augmented input space $\mathcal{G}\mathcal{X}_{aux}$, defined as $\mathcal{G}\mathcal{X}_{aux} = \{\hat{x}|\hat{x} = \lambda x_1 + (1 - \lambda)x_2; x_1, x_2 \in \mathcal{X}_{aux}, \lambda \in [0, 1]\}$, the following relation holds:*

$$\mathcal{X}_{aux} \subset \mathcal{G}\mathcal{X}_{aux}. \tag{18}$$

**Proof.** $\mathcal{X}_{aux} = \mathcal{X}_{aux}^{y_1} \cup \ldots \cup \mathcal{X}_{aux}^{y_i} \cup \ldots \cup \mathcal{X}_{aux}^{y_j} \cup \ldots \cup \mathcal{X}_{aux}^{y_n}$. Consider performing mixup to obtain a mixed outlier $\hat{x} = \lambda x_i + (1 - \lambda)x_j$, where $x_i \in \mathcal{X}_{aux}^{y_i}$, $x_j \in \mathcal{X}_{aux}^{y_j}$ and $y_i \neq y_j$. According to assumption 1, there exists $\lambda$ such that $\hat{x}$ exhibits different semantics from the original, i.e., $\hat{x} \notin \mathcal{X}_{aux}^{y_i}$ and $\hat{x} \notin \mathcal{X}_{aux}^{y_j}$. Clearly, the semantic of $\hat{x}$ is also inconsistent with other outliers in $\mathcal{X}_{aux}$. Therefore, $\hat{x} \notin \mathcal{X}_{aux}$. We define $\mathcal{X}_{mix} = \{\hat{x} \mid \hat{x} \notin \mathcal{X}_{aux}, \hat{x} = \lambda x_i + (1 - \lambda)x_j, x_i, x_j \in \mathcal{X}_{aux}\}$ to represents the input space of mixed outliers with distinct semantic to the original. Consequently, $\mathcal{G}\mathcal{X}_{aux} = \mathcal{X}_{aux} \cup \mathcal{X}_{mix}$, leading to $\mathcal{G}\mathcal{X}_{aux} \supset \mathcal{X}_{aux}$.

## B    EXPERIMENTAL DETAILS

### B.1    DETAILS OF DATASET

**Auxiliary OOD datasets.** In our research, we employ the downsampled ImageNet dataset (ImageNet $64 \times 64$) as a variant of the original ImageNet dataset, comprising 1,281,167 images with dimensions of $64\times64$ pixels and organized into 1000 distinct classes. Notably, there is overlap between some of these classes and those present in CIFAR-10 and CIFAR-100 datasets. It is important to emphasize that we abstain from utilizing any label information from this dataset, thereby regarding it as an unlabeled auxiliary OOD dataset. To augment our dataset for experiments, we apply a random cropping procedure to the $64\times64$ images, resulting in $32\times32$ pixel images with a 4-pixel padding. This operation, performed with a high probability, ensures that the resulting images are unlikely to contain objects corresponding to the ID classes, even if the original images featured such objects. Consequently, we retain a substantial quantity of OOD data for training purposes, yielding a low proportion of ID data within the auxiliary OOD dataset. For conciseness and clarity, we refer to this dataset as ImageNet-RC.

**Test OOD datasets.** To evaluate OOD performance of our model, we follow the procedure in Chen et al. (2021); Ming et al. (2022). Specifically, we employ six different natural image datasets as our OOD test datasets, while CIFAR-10 and CIFAR-100 serve as our ID test datasets. These six datasets are SVHN (Netzer et al., 2011), Textures (Cimpoi et al., 2014), Places365 (Zhou et al., 2016), LSUN (crop), LSUN (resize) (Yu et al., 2015), and iSUN (Xu et al., 2015). Below, we provide detailed information about these OOD test datasets, all of which consist of $32 \times 32$ pixel images.

**SVHN.** The SVHN dataset (Netzer et al., 2011) comprises color images of house numbers, encompassing ten different digit classes from 0 to 9. Originally, the test set contained 26,032 images. For our evaluation, we randomly select 1,000 test images from each digit class, creating a new test dataset with 10,000 images.

**Textures.** The Describable Textures Dataset (DTD) (Cimpoi et al., 2014) consists of textural images in the wild. We include the entire collection of 5,640 images for evaluation.

**Places365.** The Places365 dataset (Zhou et al., 2016) comprises a large-scale photographs depicting scenes classified into 365 scene categories. In the test set, there are 900 images per category. We randomly sample 10,000 images from the test set for our evaluation.

**LSUN (crop) and LSUN (resize).** The Large-scale Scene Understanding dataset (LSUN) (Yu et al., 2015) offers a testing set containing 10,000 images from 10 different scenes. We create two variants of this dataset, namely LSUN (crop) and LSUN (resize). LSUN (crop) is generated by randomly cropping image patches to the size of $32 \times 32$ pixels, while LSUN (resize) involves downsampling each image to the same size.

---

[4]The set of all possible combinations of data points and all possible values of $\lambda$ for mixup.

---

**Algorithm 2** ProMix: Provable Mixup Outlier (w.o. Greedy Sample)

---

**Input:** ID dataset $\mathcal{D}_{id}$, outliers dataset $\mathcal{D}_{aux}$, pool size $N'$, hyperparameter of mixup ratio $\sigma$, hyperparameter of Beta distribution $\alpha$;
**Output:** ID classifier $\hat{F}$, OOD detector $G$;
**for** *each iteration* **do**
   | // Randomly sample $N'$ outliers from $\mathcal{D}_{aux}$ to create subset $\mathcal{S}$.
   | $\mathcal{S} \leftarrow \{s_i \,|\, s_i \in \mathcal{D}_{aux}, \; i = 1, 2, ..., N'\}$;
   | // Copy $\mathcal{S}$ and random shuffle to create $\mathcal{S}'$.
   | $\mathcal{S}' \leftarrow \text{shuffle}(\mathcal{S})$;
   | // Apply mixup on $\mathcal{S}$ and $\mathcal{S}'$ to obtain the mixed outlier set $\mathcal{D}_{mix}$.
   | Sample $p \sim \text{Uniform}(0, 1)$;
   | $\mathcal{D}_{mix} \leftarrow \{\hat{x}_i \,|\, \hat{x}_i = \lambda x_i + (1 - \lambda)x'_i; \; x \in \mathcal{S}, \; x' \in \mathcal{S}'; \; i = 1, 2, ..., N';$
   | **if** $p \leq \sigma$, **then** $\lambda \sim \text{Beta}(\alpha, \alpha)$, **else** $\lambda = 1\}$ ;
   | Train $F_\theta$ for one epoch using the training objective of Eq. 13;
**end**
Build $\hat{F}$ and $G$ according to Eq. 12 and Eq. 11;

---

**iSUN.** The iSUN dataset (Xu et al., 2015) is a subset of SUN images. We incorporate the entire collection of 8,925 images from iSUN for our evaluation.

## B.2 MORE DETAILS OF Q2 RELIABILITY.

In this experiment, our objective is to examine the impact of semantic diversity of outliers on performance and subsequently verify our theory. Moreover, to eliminate the influence of sample diversity, we assess the effect of data size on performance as well. Furthermore, we aim to validate the effectiveness of our approach on a range of auxiliary outlier datasets with varying diversity levels and data sizes.

**Varying the diversity of auxiliary outliers.** To achieve auxiliary outliers with different diversity, we manipulate the number of classes present in the auxiliary outliers dataset. Specifically, we systematically control the number of classes selected from the Imagenet-RC dataset and extract a predetermined number of samples from the chosen classes. By varying the number of classes while maintaining a constant dataset size, we can effectively control the diversity of the auxiliary outliers dataset. We randomly sample classes from the original Imagenet-RC dataset, selecting percentages of $\{10\%, 25\%, 40\%, 55\%, 70\%, 85\%, 100\%\}$ of the original Imagenet-RC classes, and adjust the dataset size to be $10\%$ of the original Imagenet-RC dataset, thus generating training datasets with different degrees of diversity.

**Varying the sample size of auxiliary outliers.** To further explore the impact of sample diversity on our experimental results, we keep the number of classes constant and vary the size of the auxiliary outliers dataset. This is achieved by applying downsampling techniques, resulting in datasets with the same classes as the original Imagenet-RC dataset but with sizes of $\{10\%, 25\%, 40\%, 55\%, 70\%, 85\%, 100\%\}$ compared to the auxiliary outliers dataset. A smaller dataset size indicates reduced sample diversity.

**Methodology.** In our comparative analysis, the Vanilla method solely applies (k+1)-way regularization when using auxiliary outliers for model training. On the other hand, the ProMix method incorporates the mixup outliers operation as an extension of the Vanilla approach. To ensure fairness in our comparison, we did not employ a greedy sampling strategy. The complete pseudocode for our approach, without the use of greedy sampling, is presented in Algorithm 2.

## B.3 MORE DETAILS OF Q4 RELIABILITY.

To explore whether mixup has unique advantages compared to other data augmentation to enhance the diversity of outliers, we select different data augmentation techniques to process outliers and validate their impact on performance. Specifically, we chose semantic-invarient data augmentation methods: noise (Rifai et al., 2011), cutout (Rifai et al., 2011), and the mixup-style augmentation cutmix (Rifai et al., 2011) for comparison with our method.

Table 4: **Main results.** Comparison between our ProMix method and other competitive OOD detection methods that are trained with the same DenseNet backbone. All values are percentages and are averaged over six OOD test datasets outlined in Section 5.1. The best results are in red. Some baseline model results are sourced from Ming et al. (2022). Our method's reported performance is based on five independent training runs using different random seeds.

| $D_{in}$ | Method | FPR95 ($\downarrow$) | AUROC ($\uparrow$) | AUPR ($\uparrow$) | ID-ACC | w./w.o. $\mathcal{D}_{aux}$ | informative | diverse |
|---|---|---|---|---|---|---|---|---|
| | MSP (Hendrycks & Gimpel, 2017) | 58.98 | 90.63 | 93.18 | 94.39 | $\times$ | NA | NA |
| | ODIN (Liang et al., 2018b) | 26.55 | 94.25 | 95.34 | 94.39 | $\times$ | NA | NA |
| | Mahalanobis (Lee et al., 2018b) | 29.47 | 89.96 | 89.70 | 94.39 | $\times$ | NA | NA |
| | Energy (Wang et al., 2021a) | 28.53 | 94.39 | 95.56 | 94.39 | $\times$ | NA | NA |
| | SSD+ (Sehwag et al., 2021) | 7.22 | 98.48 | 98.59 | NA | $\times$ | NA | NA |
| CIFAR-10 | OE (Hendrycks & andThomas G. Dietterich, 2019) | 9.66 | 98.34 | 98.55 | 94.12 | $\checkmark$ | $\times$ | $\times$ |
| | SOFL (Mohseni et al., 2020) | 5.41 | 98.98 | 99.10 | 93.68 | $\checkmark$ | $\times$ | $\times$ |
| | CCU (Meinke & Hein, 2019) | 8.78 | 98.41 | 98.69 | 93.97 | $\checkmark$ | $\times$ | $\times$ |
| | Energy (w. $D_{aux}$) (Wang et al., 2021a) | 4.62 | 98.93 | 99.12 | 92.92 | $\checkmark$ | $\times$ | $\times$ |
| | NTOM (Chen et al., 2021) | $4.00 \pm 0.22$ | $99.09 \pm 0.05$ | $98.61 \pm 0.32$ | $94.26 \pm 0.11$ | $\checkmark$ | $\checkmark$ | $\times$ |
| | POEM (Ming et al., 2022) | $2.54 \pm 0.56$ | $99.40 \pm 0.05$ | $99.50 \pm 0.07$ | $93.49 \pm 0.27$ | $\checkmark$ | $\checkmark$ | $\times$ |
| | ProMix (ours) | $2.18 \pm 0.16$ | $99.43 \pm 0.02$ | $99.01 \pm 0.02$ | $94.32 \pm 0.28$ | $\checkmark$ | $\checkmark$ | $\checkmark$ |
| | MSP (Hendrycks & Gimpel, 2017) | 80.30 | 73.13 | 76.97 | 74.05 | $\times$ | NA | NA |
| | ODIN (Liang et al., 2018b) | 56.31 | 84.89 | 85.88 | 74.05 | $\times$ | NA | NA |
| | Mahalanobis (Lee et al., 2018b) | 47.89 | 85.71 | 87.15 | 74.05 | $\times$ | NA | NA |
| | Energy (Wang et al., 2021a) | 65.87 | 81.50 | 84.07 | 74.05 | $\times$ | NA | NA |
| | SSD+ (Sehwag et al., 2021) | 38.32 | 88.91 | 89.77 | NA | $\times$ | NA | NA |
| CIFAR-100 | OE (Hendrycks & andThomas G. Dietterich, 2019) | 19.54 | 94.93 | 95.26 | 74.25 | $\checkmark$ | $\times$ | $\times$ |
| | SOFL (Mohseni et al., 2020) | 19.32 | 96.32 | 96.99 | 73.93 | $\checkmark$ | $\times$ | $\times$ |
| | CCU (Meinke & Hein, 2019) | 19.27 | 95.02 | 95.41 | 74.49 | $\checkmark$ | $\times$ | $\times$ |
| | Energy (w. $D_{aux}$)($Wang\ et\ al.$, 2021a) | 19.25 | 96.68 | 97.44 | 72.39 | $\checkmark$ | $\times$ | $\times$ |
| | NTOM (Chen et al., 2021) | $18.77 \pm 0.75$ | $96.69 \pm 0.12$ | $96.49 \pm 0.33$ | $74.52 \pm 0.31$ | $\checkmark$ | $\checkmark$ | $\times$ |
| | POEM (Ming et al., 2022) | $15.14 \pm 1.16$ | $97.79 \pm 0.17$ | $98.31 \pm 0.12$ | $73.41 \pm 0.21$ | $\checkmark$ | $\checkmark$ | $\times$ |
| | ProMix (ours) | $10.37 \pm 0.43$ | $98.03 \pm 0.02$ | $98.63 \pm 0.09$ | $74.26 \pm 0.19$ | $\checkmark$ | $\checkmark$ | $\checkmark$ |

**Noise.** Here, we introduce an appropriate level of noise to the training data to augment its diversity and quantity. We incorporate Gaussian noise with a mean of 0 and a variance of 0.1. To mitigate the risk of model overfitting to Gaussian noise, wherein the model classifies any image with Gaussian noise as an outlier (OOD) and any noise-free image as an ID sample, this type of noise is applied to half of the outlier samples during the model training phase.

**Cutout.** Cutout is a data augmentation technique that introduces random masking of small regions in input images, preventing the model from relying on specific features. In our study, we apply the cutout augmentation to half of the outlier samples. This involves randomly masking out small regions within these outlier images by setting all pixel values in the masked regions to zero.

**Cutmix.** Cutmix is a data augmentation technique that involves randomly selecting two images from the training set, cropping one image, and replacing the cropped region with the corresponding region from the other image. The resulting mixed image is assigned a label that is a weighted average of the labels corresponding to the two original images. In our experimentation, we apply Cutmix data augmentation to half of the outlier samples.

The results of our study indicate that the inclusion of semantically invariant data augmentation techniques such as adding noise or applying cutout does not yield improved performance. While these methods have the potential to enhance sample diversity (data size) and theoretically improve the generalization error of the model, they fail to address the issue of distribution shift. Moreover, these techniques introduce inherent noise that deteriorates the quality of the original data, resulting in a decline in performance. Among the mixup methods investigated, Cutmix introduces noise at clipping boundaries, which can potentially lead to overfitting and inferior performance compared to the ProMix approach. As a result, our proposed method offers unique advantages in addressing these challenges.

# C ADDITIONAL RESULTS

## C.1 FULL RESULTS WITH STANDARD DEVIATION

In Tab. 4, Tab. 5 and Tab. 6, we present the experimental results for all evaluation metrics along with the corresponding standard deviations. From the experimental results we can draw similar conclusions as those in Sec. 5.

Table 5: **Ablation study on module contributions.** Performance averaged (%) over six OOD test datasets from Section 5.1. The best results are in red. Our method's reported performance is based on five independent training runs using different random seeds.

| Mixup | Greedy sample | CIFAR-10 | | | | CIFAR-100 | | | |
|---|---|---|---|---|---|---|---|---|---|
| | | FPR (↓) | AUROC (↑) | AUPR (↑) | ID-ACC | FPR (↓) | AUROC (↑) | AUPR (↑) | ID-ACC |
| × | × | $5.14 \pm 0.78$ | $98.91 \pm 0.12$ | $98.89 \pm 0.17$ | $94.40 \pm 0.15$ | $20.69 \pm 0.57$ | $95.90 \pm 0.33$ | $95.65 \pm 0.13$ | $74.27 \pm 0.17$ |
| × | ✓ | $4.00 \pm 0.22$ | $99.09 \pm 0.05$ | $98.61 \pm 0.32$ | $94.26 \pm 0.11$ | $18.77 \pm 0.75$ | $96.69 \pm 0.12$ | $96.49 \pm 0.33$ | $74.52 \pm 0.31$ |
| ✓ | × | $3.28 \pm 0.45$ | $99.23 \pm 0.05$ | $\mathbf{99.45 \pm 0.19}$ | $94.30 \pm 0.12$ | $13.33 \pm 1.20$ | $97.35 \pm 0.19$ | $\mathbf{98.66 \pm 0.18}$ | $74.28 \pm 0.20$ |
| ✓ | ✓ | $\mathbf{2.18 \pm 0.16}$ | $\mathbf{99.43 \pm 0.02}$ | $99.01 \pm 0.02$ | $94.32 \pm 0.28$ | $\mathbf{10.37 \pm 0.43}$ | $\mathbf{98.03 \pm 0.02}$ | $98.63 \pm 0.09$ | $74.26 \pm 0.19$ |

Table 6: **Ablation on data augmentation method.** Averaged performance (%) over six OOD test datasets from Section 5.1. The best results are in red. Our method's reported performance is based on five independent training runs using different random seeds.

| | Semantic change ? | CIFAR-10 | | | | CIFAR-100 | | | |
|---|---|---|---|---|---|---|---|---|---|
| | | FPR (↓) | AUROC (↑) | AUPR (↑) | ID-ACC | FPR (↓) | AUROC (↑) | AUPR (↑) | ID-ACC |
| Noise | × | $7.84 \pm 0.81$ | $98.56 \pm 0.13$ | $98.87 \pm 0.18$ | $94.02 \pm 0.15$ | $26.01 \pm 1.09$ | $94.87 \pm 0.13$ | $96.06 \pm 0.13$ | $74.69 \pm 0.30$ |
| Cutout | × | $8.83 \pm 1.89$ | $98.45 \pm 0.31$ | $98.37 \pm 0.36$ | $94.27 \pm 0.04$ | $24.36 \pm 0.22$ | $95.01 \pm 0.33$ | $94.91 \pm 0.61$ | $74.55 \pm 0.19$ |
| Vanilla | NA | $5.41 \pm 0.78$ | $98.91 \pm 0.12$ | $98.89 \pm 0.17$ | $94.40 \pm 0.15$ | $20.69 \pm 0.57$ | $95.90 \pm 0.09$ | $95.65 \pm 0.13$ | $74.27 \pm 0.17$ |
| Cutmix | ✓ | $5.76 \pm 0.78$ | $98.95 \pm 0.11$ | $99.08 \pm 0.27$ | $94.37 \pm 0.10$ | $21.23 \pm 1.27$ | $96.03 \pm 0.32$ | $96.34 \pm 0.45$ | $74.45 \pm 0.29$ |
| Mixup (ours) | ✓ | $\mathbf{3.28 \pm 0.45}$ | $\mathbf{99.23 \pm 0.05}$ | $\mathbf{99.45 \pm 0.19}$ | $94.30 \pm 0.12$ | $\mathbf{13.33 \pm 1.20}$ | $\mathbf{97.35 \pm 0.19}$ | $\mathbf{98.66 \pm 0.18}$ | $74.28 \pm 0.20$ |

Table 7: **main results on individual OOD dataset.** We provide the results of ProMix on each OOD dataset from Section 5.1. The best results are in red. Our method's reported performance is based on five independent training runs using different random seeds.

| OOD dataset | CIFAR-10 | | | | CIFAR-100 | | | |
|---|---|---|---|---|---|---|---|---|
| | FPR (↓) | AUROC (↑) | AUPR (↑) | ID-ACC | FPR (↓) | AUROC (↑) | AUPR (↑) | ID-ACC |
| LSUN (crop) | $6.23 \pm 0.73$ | $98.68 \pm 0.12$ | $98.44 \pm 0.16$ | $94.32 \pm 0.28$ | $26.18 \pm 26.18$ | $95.76 \pm 0.21$ | $94.79 \pm 0.31$ | $74.26 \pm 0.19$ |
| LSUN (resize) | $0.00 \pm 0.00$ | $100.00 \pm 0.00$ | $100.00 \pm 0.00$ | $94.32 \pm 0.28$ | $0.00 \pm 0.00$ | $99.99 \pm 0.01$ | $99.98 \pm 0.02$ | $74.26 \pm 0.19$ |
| iSUN | $0.00 \pm 0.00$ | $100.00 \pm 0.00$ | $100.00 \pm 0.00$ | $94.32 \pm 0.28$ | $0.03 \pm 0.01$ | $99.97 \pm 0.01$ | $100.00 \pm 0.00$ | $74.26 \pm 0.19$ |
| dtd | $0.81 \pm 0.06$ | $99.70 \pm 0.02$ | $99.99 \pm 0.00$ | $94.32 \pm 0.28$ | $4.59 \pm 0.44$ | $98.76 \pm 0.09$ | $99.98 \pm 0.01$ | $74.26 \pm 0.19$ |
| places365 | $5.01 \pm 0.33$ | $98.74 \pm 0.09$ | $96.46 \pm 0.23$ | $94.32 \pm 0.23$ | $21.33 \pm 0.29$ | $95.83 \pm 0.16$ | $99.79 \pm 0.09$ | $74.26 \pm 0.19$ |
| SVHN | $1.06 \pm 0.29$ | $99.44 \pm 0.09$ | $99.18 \pm 0.14$ | $94.32 \pm 0.28$ | $10.07 \pm 1.69$ | $97.88 \pm 0.16$ | $97.25 \pm 0.27$ | $74.26 \pm 0.19$ |
| average | $2.18 \pm 0.16$ | $99.43 \pm 0.02$ | $99.01 \pm 0.02$ | $94.32 \pm 0.28$ | $10.37 \pm 0.43$ | $98.03 \pm 0.02$ | $98.63 \pm 0.09$ | $74.26 \pm 0.19$ |

Table 8: **Hyperparameter Analysis.** Comparison results with varying hyperparameter $\alpha \in \{0.5, 0.75, 1.0, 1.25, 1.5\}$. The best results are in red. All values are percentages and are averaged over six OOD test datasets described in Sec. 5.1.

| $\alpha$ | CIFAR-10 | | | | CIFAR-100 | | | |
|---|---|---|---|---|---|---|---|---|
| | FPR (↓) | AUROC (↑) | AUPR (↑) | ID-ACC | FPR (↓) | AUROC (↑) | AUPR (↑) | ID-ACC |
| 0.5 | $\mathbf{1.92}$ | $\mathbf{99.48}$ | $\mathbf{99.73}$ | 94.21 | 9.76 | $\mathbf{98.13}$ | 98.99 | 74.68 |
| 0.75 | 2.27 | 99.40 | 99.60 | 94.22 | 11.19 | 97.80 | 98.67 | 73.80 |
| 1.0 | 2.18 | 99.43 | 99.01 | 94.32 | 10.37 | 98.03 | 98.63 | 74.26 |
| 1.25 | 2.33 | 99.38 | 99.67 | 94.22 | $\mathbf{9.25}$ | 98.07 | $\mathbf{99.09}$ | 74.27 |
| 1.5 | 2.42 | 99.38 | 99.65 | 94.08 | 11.45 | 97.70 | 98.49 | 74.40 |

## C.2  RESULTS ON INDIVIDUAL OOD DATASET

We also provide the performance of our method on individual OOD dataset in tabel 7.

## C.3  HYPERPARAMETER ANALYSIS.

**Analysis of hyperparameter $\alpha$.** This study aims to investigate the influence of the hyperparameter $\alpha$ in the mixup technique on the performance of our proposed method. A systematic exploration was conducted by varying the value of $\alpha$ while keeping other parameters fixed. The outcomes of this analysis are summarized in a concise manner in Table 8. Remarkably, our approach demonstrated robustness to changes in the $\alpha$ hyperparameter, encompassing a broad range from 0.5 to 1.5.

**Analysis of hyperparameter $\sigma$.** Furthermore, an extensive investigation was performed on the CIFAR-10 and CIFAR-100 datasets to assess the impact of the hyperparameter $\sigma$ in the mixup technique. The fine-tuning of $\sigma$ was carried out while maintaining the other parameters at their original

Table 9: **Hyperparameter Analysis.** Comparison results with varying hyperparameter $\sigma \in \{0, 0.25, 0.5, 0.75, 1.0\}$. The best results are in red. All values are percentages and are averaged over six OOD test datasets described in Sec. 5.1.

| $\sigma$ | CIFAR-10 | | | | CIFAR-100 | | | |
|---|---|---|---|---|---|---|---|---|
| | FPR ($\downarrow$) | AUROC ($\uparrow$) | AUPR ($\uparrow$) | ID-ACC | FPR ($\downarrow$) | AUROC ($\uparrow$) | AUPR ($\uparrow$) | ID-ACC |
| 0 | 4.00 | 99.09 | 98.61 | 94.26 | 18.77 | 96.69 | 96.69 | 74.52 |
| 0.25 | 2.39 | 99.37 | 98.74 | 94.20 | 15.74 | 97.17 | **98.92** | 74.25 |
| 0.5 | **2.18** | **99.43** | 99.01 | 94.32 | 10.37 | **98.03** | 98.63 | 74.26 |
| 0.75 | 2.26 | 99.35 | 99.15 | 94.34 | 11.70 | 97.84 | 98.45 | 74.30 |
| 1.0 | 2.46 | 99.38 | **99.62** | 94.33 | **10.03** | 97.94 | 98.89 | 74.41 |

Table 10: **OOD Score Analysis.** Comparison results with different OOD scores. All values are percentages and are averaged over six OOD test datasets described in Sec. 5.1.

| Method | CIFAR-10 | | | | CIFAR-100 | | | |
|---|---|---|---|---|---|---|---|---|
| | FPR ($\downarrow$) | AUROC ($\uparrow$) | AUPR ($\uparrow$) | ID-ACC | FPR ($\downarrow$) | AUROC ($\uparrow$) | AUPR ($\uparrow$) | ID-ACC |
| Energy | $3.42 \pm 0.55$ | $99.15 \pm 0.10$ | $99.11 \pm 0.08$ | $93.25 \pm 0.03$ | $19.02 \pm 1.01$ | $96.44 \pm 0.19$ | $96.42 \pm 0.22$ | $72.61 \pm 0.33$ |
| Energy + ProMix | $2.01 \pm 0.58$ | $99.41 \pm 0.10$ | $99.53 \pm 0.07$ | $93.11 \pm 0.07$ | $12.74 \pm 1.08$ | $97.58 \pm 0.11$ | $97.99 \pm 0.18$ | $72.56 \pm 0.49$ |
| OE | $9.14 \pm 0.64$ | $98.43 \pm 0.07$ | $98.60 \pm 0.22$ | $94.09 \pm 0.07$ | $19.97 \pm 1.27$ | $94.89 \pm 0.32$ | $96.18 \pm 0.86$ | $74.21 \pm 0.07$ |
| OE + ProMix | $8.09 \pm 0.28$ | $98.54 \pm 0.07$ | $98.88 \pm 0.10$ | $94.01 \pm 0.13$ | $16.43 \pm 0.94$ | $95.72 \pm 0.17$ | $98.08 \pm 0.15$ | $74.35 \pm 0.36$ |

Table 11: **Main results on large-scale datasets.** Comparison results with different OOD scores.

| Method | SSB-hard | | NINCO | | iNaturalist | | Textures | | OpenImage-O | | Average | |
|---|---|---|---|---|---|---|---|---|---|---|---|---|
| | FPR ($\downarrow$) | AUROC ($\uparrow$) | FPR ($\downarrow$) | AUROC ($\uparrow$) | FPR ($\downarrow$) | AUROC ($\uparrow$) | FPR ($\downarrow$) | AUROC ($\uparrow$) | FPR ($\downarrow$) | AUROC ($\uparrow$) | FPR ($\downarrow$) | AUROC ($\uparrow$) |
| MSP | 78.90 | 80.24 | 76.25 | 81.11 | 67.64 | 81.97 | 76.12 | 82.77 | 78.19 | 80.90 | 75.42 | 81.40 |
| Energy | 73.56 | 84.12 | 68.86 | 84.91 | 52.55 | 86.78 | 67.68 | 87.66 | 72.76 | 84.18 | 67.08 | 85.53 |
| OE | 67.78 | 84.85 | 65.07 | 85.81 | 51.31 | 88.92 | 65.72 | 86.64 | 62.75 | 86.71 | 62.53 | 86.59 |
| K+1 | 77.48 | 82.31 | 73.14 | 84.61 | 71.95 | 84.47 | 91.49 | 78.87 | 75.64 | 83.59 | 77.94 | 82.77 |
| ProMix (OE) | 42.80 | 90.82 | 36.75 | 92.19 | 35.41 | 92.12 | 25.86 | 95.12 | 33.48 | 93.43 | 34.86 | 92.74 |
| ProMix (K+1) | 50.37 | 90.42 | 43.99 | 91.76 | 55.62 | 88.91 | 45.30 | 92.33 | 40.31 | 92.80 | 47.12 | 91.24 |

predefined values. The corresponding experimental results are presented in Table 9. Notably, incorporating mixup on a certain fraction of outliers substantially improved the model's performance, thereby confirming the effectiveness of our proposed approach.

## C.4 ABLATION STUDY ON DIFFERENT OOD SCORE.

The form of the OOD score adopted in this paper, derived as classification confidence for the $K + 1$ class, is a specific case in implementation. We conduct experiments with more general forms of anomaly scores and provide the corresponding experimental results. In particular, we chose two distinct anomaly scores, energy and OE, for the application of our method. The experimental results are outlined in table 10. The experimental results demonstrate that our method is effective in broader settings, accommodating arbitrary OOD scores.

## C.5 EXPERIMENTS ON LARGE-SCALE DATASETS

We evaluate our approach on larger-scale benchmarks and provide the corresponding experiments under the setting in Yang et al. (2022). Our experimental setup involved utilizing 200 classes from ImageNet-1k (224x224 resolution) as the ID class and an additional 800 classes as outliers. We conducted tests across five different out-of-distribution OOD test sets. Specifically, SSB-hard and NINCO were included as part of the near-OOD group for ImageNet-1k, while iNaturalist, Textures, and OpenImage-O were classified under the far-OOD group. The experimental results are represented in table 11:

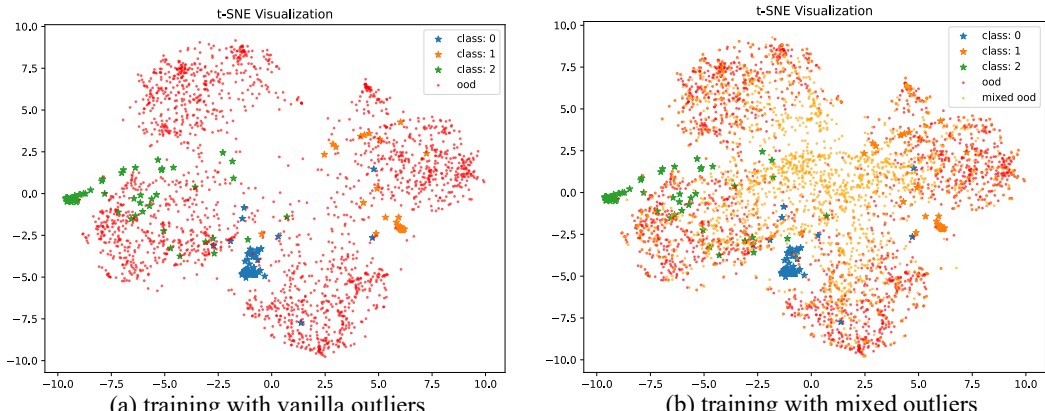

(a) training with vanilla outliers       (b) training with mixed outliers

Figure 3: 2D Visualization of data representations in the DNN's feature space. Stars of different colors represent data from distinct ID categories, while dots represent outliers. Red dots signify vanilla outliers, and orange dots denote mixed outliers generated through mixup. In (a), outliers are dispersed and form clustered distributions. In (b), mixup applied to outliers generates mixed outliers, effectively bridging the sparsity observed among vanilla outliers and covering a more extensive area, thereby enhancing the diversity of outliers.

### C.6 VISUALIZATION

The utilization of mixup techniques has the potential to extend the high-dimensional spatial coverage of auxiliary outlier data, thereby enriching its diversity. Auxiliary outliers encompass data from diverse classes, indicating significant semantic distinctions among different categories. This implies that, in a high-dimensional space, auxiliary outliers are clustered by category, with sparse distribution between outliers from distinct categories. Mixup operates by interpolating between outliers with different semantic information belonging to distinct clusters, effectively bridging the sparsity between these clusters. This process empowers auxiliary outliers to cover a broader region in high-dimensional space, thereby enhancing the diversity of auxiliary outliers. To provide a clearer understanding of why mixup enhances the diversity of auxiliary outliers, visualizations have been included in Figure 3.

## D HARDWARE AND SOFTWARE

We run all the experiments on NVIDIA GeForce RTX 3090 GPU. Our implementations are based on Ubuntu Linux 18.04 with Python 3.8.

