# OpenReview forum: "Guaranteed Out-Of-Distribution Detection with Diverse Auxiliary Set"
_ICLR.cc/2024/Conference — Submitted to ICLR 2024_

### Official Review · Reviewer_iVGn · 2023-10-26

**Soundness:** 4 excellent
**Presentation:** 4 excellent
**Contribution:** 2 fair
**Rating:** 8
**Confidence:** 4

**Summary:**

The manuscript proposes a new method for improving out-of-distribution detection in the presence of a finite set of auxiliary negative samples. The manuscript bounds generalization error using empirical error, reducible error, and distribution shift error. The latter is caused by the finite auxiliary dataset which cannot represent all possible test outliers. Therefore, the manuscript proposes an augmentation technique based on mixup which increases the variety of auxiliary data.  In practice, negative training samples are a convex combination of different auxiliary negatives. During inference,
 OOD samples are detected based on the classification confidence of K+1st class.
The proposed method achieves competitive results on relevant benchmarks for OOD detection.

**Strengths:**

S1. The method is well motivated by extensive and sound theoretical analysis. Proofs are mostly easy to follow and seem correct.

S2. The proposed augmentation technique for auxiliary datasets yields competitive results.

**Weaknesses:**

W1. The presented method relies on anomaly score derived as classification confidence for K+1 class. Can the method work in more general settings with arbitrary anomaly scores (such as entropy or energy)? Demonstrating robustness to various anomaly scores would strengthen manuscript contributions.

W2. Mixup-based augmentation techniques are already considered in [a]. The manuscript should clearly outline the differences.

W3. Missing related works which replace auxiliary negatives with properly trained generative models [b,c,d].

[a] Dan Hendrycks, Andy Zou, Mantas Mazeika, Leonard Tang, Bo Li, Dawn Song, Jacob Steinhardt: PixMix: Dreamlike Pictures Comprehensively Improve Safety Measures. CVPR 2022

[b] Kimin Lee, Honglak Lee, Kibok Lee, Jinwoo Shin: Training Confidence-calibrated Classifiers for Detecting Out-of-Distribution Samples. ICLR 2018

[c] Matej Grcic, Petra Bevandic, Sinisa Segvic: Dense Open-set Recognition with Synthetic Outliers Generated by Real NVP. VISIGRAPP 2021

[d] Shu Kong, Deva Ramanan: OpenGAN: Open-Set Recognition via Open Data Generation. ICCV 2021

**Questions:**

Q1. The error caused by the finite auxiliary negative dataset which cannot cover all possible test outliers is termed distribution shift error.  I suggest renaming the distribution shift error to outlier coverage error.

Q2. Should the first term of Eq. 13 have D_inlier instead of D_aux?

Q3. Proof of Theorem 2, could you elaborate on the equality in the line above the Eq. 16?

Q4. Misspelled "Theorm 3" below Eq. 10 in the main text.

---

> ### Author Response · Authors · 2023-11-19
>
> We sincerely thank  you for spending time and providing valuable feedback. We  appreciate all of your suggestions and we have addressed all your  questions below by providing our responses as well as our additional  experimental results.
> ## W1. More general setting with arbitrary anomaly scores.
> R1. The form of the anomaly score adopted in this paper, derived as classification confidence for the K+1 class, is a specific case in implementation. We have conducted experiments with more general forms of anomaly scores and provide the corresponding experimental results.  In particular, we chose two distinct anomaly scores, energy and OE, for the application of our method. The experimental results are outlined below:
>
> | Dataset   | method | FPR $(\downarrow)$    | AUROC $(\uparrow)$  | AUPR_out $(\uparrow)$| ID-acc |
> | --------- | ------ | ------ | ------ | ----------- | -------|
> | CIFAR-10 | energy | 3.42 ± 0.55  | 99.15 ± 0.10 | 99.11 ± 0.08 | 93.25 ± 0.03 |
> | CIFAR-10 | energy + promix | **2.01 ± 0.58** | **99.41 ± 0.10** | **99.53 ± 0.07** | 93.11 ± 0.07 |
> | CIFAR-10 | OE | 9.14 ± 0.64  | 98.43 ± 0.07 | 98.60 ± 0.22 | 94.09 ± 0.07 |
> | CIFAR-10 | OE + promix | 8.09 ± 0.28  | 98.54 ± 0.07 | 98.88 ± 0.10 | 94.01 ± 0.13 |
> | CIFAR-100 | energy | 19.02 ± 1.01 | 96.44 ± 0.19 | 96.42 ± 0.22 | 72.61 ± 0.33 |
> | CIFAR-100 | energy + promix |**12.74 ± 1.08** | **97.58 ± 0.11** | 97.99 ± 0.18 | 72.56 ± 0.49 |
> | CIFAR-100 | OE |  19.97 ± 1.27 | 94.89 ± 0.32 | 96.18 ± 0.86 | 74.21 ± 0.07 |
> | CIFAR-100 | OE + promix | 16.43 ± 0.94 | 95.72 ± 0.17 | **98.08 ± 0.15** | 74.35 ± 0.36 |
>
> The experimental results demonstrate that our method is effective in broader settings, accommodating different  widely-used anomaly scores.
>
> ## W2. Differences with PixMix [a]
> **R2.** PixMix stands out as an elegant and  effective data augmentation technique, introducing new complexity information into the training procedure through the utilization of fractals and feature visualizations, which comprehensively improves safety measures. However, there are several key differences between this approach and ours:
> (1) **Different Motivations:** Pixmix approach is introduced for multiple important purposes, e.g., enhancing OOD  robustness, ensuring prediction consistency, enhancing resilience  against adversaries, obtaining well-calibrated uncertainty estimates, and facilitating the detection of anomalous inputs. Our work focuses on theoretical analysis to show why and how the diversity of outliers impacts the OOD detection ability. As a result, we  apply mixup to increase the diversity of outliers and thus improve OOD detection capabilities.
> (2) **Different Mixup Goal:** The main role of mixup in Promix is to fuse the original images with complex fractals and feature visualizations, as fractals and feature visualizations do not belong to any particular class, this method does not change the semantics of the original data. However, mixup in our method is to combine outliers with different semantic information to generate new outliers, and the new outliers have different semantic information with the original data. Moreover, unlike pixmix, we do not perform any operations on the ID data, only on outliers.
>
> ## W3. Missing related works which replace auxiliary negatives with properly trained generative models.
> **R3:** We thank the reviewer for pointing out these exciting works that use generative models to synthesize outliers for training OOD detectors. [b] is the first to propose using a generative model to generate outliers on  the classification boundary of ID data for training. [c] obtains the the synthetic outliers by sampling a generative model based on normalized flow that is trained alongside a dense discriminative model in order to produce samples at the border of the training distribution, which produces   better performance. [d] Building a discriminator on features computed by a closed-world  K-way network and carefully selected The GAN discriminators on some real outlier data, augmenting the outlier set with adversarially synthesized "fake"  data, which achieves good performance in open-set recognition. We will review these works in the manuscript.
>
> [a] Dan Hendrycks, Andy Zou, Mantas Mazeika, Leonard Tang, Bo Li, Dawn Song, Jacob Steinhardt: PixMix: Dreamlike Pictures Comprehensively Improve Safety Measures. CVPR 2022
>
> [b] Kimin Lee, Honglak Lee, Kibok Lee, Jinwoo Shin: Training  Confidence-calibrated Classifiers for Detecting Out-of-Distribution  Samples. ICLR 2018
>
> [c] Matej Grcic, Petra Bevandic, Sinisa Segvic: Dense Open-set  Recognition with Synthetic Outliers Generated by Real NVP. VISIGRAPP  2021
>
> [d] Shu Kong, Deva Ramanan: OpenGAN: Open-Set Recognition via Open Data Generation. ICCV 2021

---

> > ### Author Response · Authors · 2023-11-19
> >
> > ## Response for Q1,Q2,Q3,Q4
> > **Response.** Thank you for suggesting the name change from "distribution shift error" to "outlier coverage error" (Q1). The suggested term more precisely conveys the error's source - the incomplete coverage of potential outliers by the finite auxiliary dataset. This enhances the clarity of our paper for readers, and I appreciate your insightful suggestion to improve this crucial concept. Regarding the error in Eq. 13, thank you for pointing it out. We have updated D_aux to D_id (Q2). For the clarification on the equality above Eq. 16 in the proof of Theorem 2 (Q3), there was a notation error, which has now been corrected. The accurate representation is as follows:
> >
> > $\text{max} ( \underset{h\in \mathcal H^*_{div}}{sup}\epsilon_{x\sim\mathcal P_{\mathcal X}}(h,h^*_{ood}),\underset{h\in \mathcal H^*_{aux}-\mathcal H^*_{div}}{sup}\epsilon_{x\sim\mathcal P_{\mathcal X}}(h,h^*_{ood}) )
> > =\underset{h\in\mathcal H^*_{aux}}{sup}\epsilon_{x\sim\mathcal P_{\mathcal X}}(h,h^*_{ood})$
> >
> > Finally, I appreciate you catching the misspelling of "Theorem 3" below Eq. 10 in the main text. It has been corrected (Q4).

---

> > > ### Comment · Reviewer_iVGn · 2023-11-23
> > > **Post rebuttal**
> > >
> > > The authors successfully addressed all of my concerns. The manuscript presents an interesting theoretical analysis which leads to practical performance improvements. Even though similar methods already exist, the manuscript includes more comprehensive reasoning behind these approaches. Finally, the response to (pfu4) presents good results on challenging large-scale experiments. Therefore, I vote for the acceptance of the paper and increase my score.

---

> > > > ### Author Response · Authors · 2023-11-23
> > > > **Thanks for raising your score!**
> > > >
> > > > We deeply appreciate you dedicating time and effort to thoroughly review our work. Your insightful comments significantly improved our paper. We sincerely gratitude your constructive feedback and guidance in enhancing our work. Thank you.

---

### Official Review · Reviewer_aAfD · 2023-10-28

**Soundness:** 3 good
**Presentation:** 2 fair
**Contribution:** 2 fair
**Rating:** 5
**Confidence:** 4

**Summary:**

Constrained by limited access to auxiliary outliers and the high cost of data collection, the authors propose Provable Mixup Outlier (ProMix), a simple yet practical approach that utilizes mixup to enhance auxiliary outlier diversity. By training with these diverse outliers, the proposed method achieves superior OOD detection. The authors also provide insightful theoretical analysis to verify that the proposed method achieves better performance than prior works.

**Strengths:**

1. The paper is written well and is easy to understand.
2. The studied problem is very important.
3. The results seem to outperform state-of-the-art.

**Weaknesses:**

1. The authors are suggested to be more careful and rigorous with the theoretical terms used in the paper, such as the "generalization risk" is not rigorous in theory.
2. What are the assumptions made in theory? It is better to add more discussions on them and the validity of making the assumptions.
3. For theorem 2, did the authors consider the sample complexity of both the vanilla auxiliary outlier set and the diverse outlier set? The sample complexity and model complexity should play an important role in the bound during analysis for these two cases.
4. For theorem 3, I am curious how well the h_mix compared with the predictor h_div.
5. It is still not intuitively making sense to me why does mixup can create diverse outlier examples. It will not change the data distribution much since it is only doing the interpolation work. What if the auxiliary outlier set is very constrained in the convex high-dimensional input space, would mixup increase the diversity of the outlier set significantly?
6. If possible, the authors are encouraged to provide empirical evidence on large-scale benchmarks. The current setting does not seem to suggest the diversity will create a performance improvement significantly.

**Questions:**

see above

---

> ### Author Response · Authors · 2023-11-19
>
> We sincerely thank you for all the comments, we have addressed all your questions below and  hope they have clarified all confusion you had about our work.
>
> # W1.  more careful and rigorous with the theoretical terms , such as the "generalization risk" is not rigorous in theory.
> **R1.** Thank you for the important suggestion. We have changed "generalization risk" to "generalization error."
>
> # W2.  Discussion about the assumptions made in theory.
> **R2.** The key assumptions made in the theoretical analysis are:
> (1) Semantic difference under mixup: This assumption states that when two samples belonging to different classes are mixed, the resulting mixed sample may have a semantic meaning different from the original samples. This assumption is grounded in the observation that mixed samples often presents an ambiguous semantic information. The practice of using 'soft labels' in vanilla mixup also supports this assumption, suggesting that the mixup process alters the original semantics of the samples.
> (2) Low error of the ideal OOD detector $\lambda$: This assumes we can find an ideal detector in the hypothesis class H that achieves a sufficient small or near zero error on the OOD distribution. With powerful deep networks, we can expect to find detectors with small $\lambda$. As if $\lambda$ is large, there does not exists a good OOD detectors in the hypothesis classes, which implies that we cannot expect to achieve good OOD detection.
> (3) Equal empirical error when training with more diverse outliers: The assumption is rooted in the over-parameterized setting of deep neural networks. The abundance of trainable model parameters enables these models to achieve near-perfect fitting to a wide range of functions during training, resulting in sufficiently small training losses. This implies that, when well-trained, the model can attain comparable and low empirical errors when exposed to more diverse outliers.
>
> # W3. For theorem 2, did the authors consider the sample complexity of both the vanilla auxiliary outlier set and the diverse outlier set? The sample complexity and model complexity should play an important role in the bound during analysis for these two cases.
>
> **R3.** Thank you for raising questions regarding the complexity of samples and models. In our research, our primary focus lies in addressing the errors induced by distribution shift caused by auxiliary outliers that fail to fully cover the complexity of the real-world out-of-distribution (OOD) scenarios. This limitation stands as a critical factor constraining the performance of OOD detection. Issues related to generalization errors arising from model and sample complexities are more prevalent in traditional machine learning. However, in our specific task, such errors are often overshadowed by the disparities introduced by auxiliary outliers' inability to adequately cover the complexity of real-world OOD data. Therefore, in the context of our paper, we assume the model to be over-parameterized and the training data to be abundant, implying that the model can effectively fit the distribution of the training data [1]. We acknowledge your perspective, we will give a more comprehensive and precise analysis about how sample complexity and model complexity jointly influence the ultimate decision boundaries in future work.
>
> # W4. how well the h_mix compared with the predictor h_div.
>
> **R4.** We conducted experiments to demonstrate the performance comparison between $h_{mix}$, trained with mixed outliers, and $h_{div}$, trained with more diverse outliers. Specifically, we chose CIFAR-100 as the in-domain (ID) dataset. For $h_{mix}$, we extracted data from 250 classes in ImageNet-RC to serve as less diverse outliers, and applied mixup outliers during training. As for $h_{div}$, we utilized the entire set of classes (1000 classes) from ImageNet-RC as more diverse outliers for direct training. Additionally, we trained model $h$ by directly training on data from 250 classes as outliers from ImageNet-RC. The experimental results are as follows:
>
> | model | outlier dataset | FPR $(\downarrow)$ | AUROC $(\uparrow)$ | AUPR_out $(\uparrow)$ | ID-acc $(\uparrow)$ |
> | ---- | ---- | ---- | ---- | ----  |---- |
> | $h$ | imagenet-RC (250 classes) | 23.01 ± 3.23 | 95.69 ± 0.53 | 96.26 ± 0.79 | 74.49 ± 0.28 |
> | $h_{div} $ | imagenet-RC (1000 classes)| 18.82 ± 1.40 | 96.42 ± 0.28 | 96.45 ± 0.49 | 74.21 ± 0.25 |
> | $h_{mix}$ | imagenet-RC (250 classes) | 18.25 ± 1.81 | 96.11 ± 0.35 | 98.24 ± 0.38 | 74.24 ± 0.33 |
>
> From the experiments, it is evident that the model $h_{div}$, trained with more diverse outliers, outperforms the model $h$ trained without such diversity. However, after enhancing the diversity of outliers through mixup, the performance of $h_{mix}$ surpasses that of $h_{div}$. This validates the effectiveness of our approach and the underlying theoretical framework. The more intuitive comparison is illustrated in Figure 2(a) of our paper.

---

> > ### Author Response · Authors · 2023-11-19
> >
> > # W5. why does mixup can create diverse outlier examples? What if the auxiliary outlier set is very constrained in the convex high-dimensional input space, would mixup increase the diversity of the outlier set significantly?
> >
> > **R5.** Thanks for the questions.
> > (1) To understand why mixup can create diverse outlier examples, please note that: (1.1) The interpolation process of mixup brings about a significant alteration in the data distribution. Unlike traditional data augmentation methods, such as adding noise or rotation, which maintain semantic consistency with the original data, mixup operates as an out-of-manifold data augmentation method [2]. The soft labels used in mixup introduce probabilistic descriptions of similarity between the newly generated samples and the original class, resulting in samples that exhibit semantic inconsistency with the original data. Consequently, mixup-generated samples deviate from the original data manifold, leading to a notable change in data distribution. Considering our definition of diversity, these out-of-manifold samples generated by mixup extend diversity. (1.2) Mixup outliers can expand the high dimensional spatial coverage of auxiliary outlier data and enhance its diversity. Auxiliary outliers comprise data from diverse classes, indicating significant semantic distinctions among different categories, which implies that, in high-dimensional space, auxiliary outliers are clustered by category, with sparse distribution between outliers from distinct categories. Mixup interpolates between outliers with different semantic information belonging to distinct clusters, bridging the sparsity between these clusters. This process enables auxiliary outliers to cover a broader region in high-dimensional space, which means that mixup enhances the diversity of auxiliary outliers. (1.3) To provide a clearer understanding of this issue, we have included visualizations in Appendix, illustrating in the feature space why mixup can create diverse outlier examples.
> > (2) Actually, auxiliary datasets often exhibit sparsity in high-dimensional space and not constrained to a specific region. Assuming that in the worst case, auxiliary outliers are highly constrained within a convex high-dimensional input space, with all outliers being highly similar and belonging to the same class, mixup may not significantly increase data diversity. However, it is important to note that mixup does not adversely impact the model's performance under such conditions.
> >
> > # W6. empirical evidence on large-scale benchmarks.
> >
> > **R6.** We appreciate the suggestion to evaluate our approach on larger-scale benchmarks and provide the corresponding experiments under the setting in [3]. Our experimental setup involved utilizing 200 classes from ImageNet-1k (224x224 resolution) as the in-distribution (ID) class and an additional 800 classes as outliers. We conducted tests across five different out-of-distribution (OOD) test sets. Specifically, 'SSB-hard' and 'NINCO' were included as part of the near-OOD group for ImageNet-1k, while 'iNaturalist,' 'Textures,' and 'OpenImage-O' were classified under the far-OOD group. The experimental results are as follows:
> >
> > | FPR95 ($\downarrow$) / AUROC ($\uparrow$) | NINCO | SSB_hard | Textures | iNaturalist | OpenImage-O | average |
> > |--- | ----- | -------- | ---   | ----------- | --------- | ------- |
> > |MSP | 78.90 / 80.24 | 76.25 / 81.11 | 67.64 / 81.97 | 76.12 / 82.77 | 78.19 / 80.90 | 75.42 / 81.40 |
> > | energy (w.o. outliers) | 73.56 / 84.12 | 68.86 / 84.91 | 52.55 / 86.78 | 67.68 / 87.66 | 72.76 / 84.18 | 67.08 / 85.53 |
> > | k+1 | 77.48 / 82.31 | 73.14 / 84.61 | 71.95 / 84.47 | 91.49 / 78.87 | 75.64 / 83.59 | 77.94 / 82.77 |
> > | OE | 67.78 / 84.85 | 65.07 / 85.81 | 51.31 / 88.92 | 65.72 / 86.64 | 62.75 / 86.71 | 62.53 / 86.59 |
> > | promix (k+1) | 50.37 / 90.42 | 43.99 / 91.76 | 55.62 / 88.91 | 45.30 / 92.33 | 40.31 / 92.80 | 47.12 / 91.24 |
> > | promix (OE) | **42.80 / 90.82** | **36.75 / 92.19** | **35.41 / 92.12** | **25.86 / 95.12** | **33.48 / 93.43** | **34.86 / 92.74** |
> >
> > Among them, the MSP and energy methods were trained without using outliers, while the k+1 and OE methods were trained with outliers. To validate the versatility of our method, we adapted it to the OOD scores of k+1 and OE, denoted as promix(k+1) and promix(OE), respectively. The experimental results show that our method is naturally effective on large-scale datasets both on near OOD detection and far OOD detection. The large-scale setting provides a powerful validation that the diversity will create a performance improvement significantly.
> >
> > [1] Reconciling modern machine-learning practice and the classical bias–variance trade-off.
> >
> > [2] MixUp as Locally Linear Out-Of-Manifold Regularization
> >
> > [3] OpenOOD v1.5: Enhanced Benchmark for Out-of-Distribution Detection

---

### Official Review · Reviewer_pfu4 · 2023-10-30

**Soundness:** 2 fair
**Presentation:** 3 good
**Contribution:** 2 fair
**Rating:** 5
**Confidence:** 3

**Summary:**

This work aims to address the challenges in OOD detection, particularly the limitation of current detectors to generalize from the distribution of auxiliary outliers. The authors introduce Provable Mixup Outlier (ProMix), using Mixup to increase the diversity of auxiliary outliers, with insightful theoretical analysis, leading to enhanced OOD detection. Evaluations on benchmarks like CIFAR-10 and CIFAR-100 indicate improved performance over existing techniques.

**Strengths:**

1. The paper is well-structured, making it reader-friendly and easy to follow.

2. The theoretical analysis constructed for Mixup is both enlightening and insightful.

3. Extensive ablation studies were conducted, showcasing various experiment setups and results.

**Weaknesses:**

1, Novelty Concerns: Applying mixup for OOD detection is not new. Many other works have explored this concept previously [1][2][3].

2. The minor modification the authors propose to the original Mixup (explicitly using the existing model and selecting Mixup outliers with lower OOD scores) lacks clarity in its effectiveness. There doesn't seem to be theoretical justification or ablation studies to validate the efficacy of the modification.

3. The auxiliary dataset used for training is a downsampled version of ImageNet. Compared to the utilized ID datasets CIFAR-10 and CIFAR-100, its diversity seems significantly higher. Therefore, the OOD detection performance gains achieved with such an auxiliary dataset might not be wholly convincing, even though the methods authors compared against also adopt this approach.

4. I’m a little concerned about the performance of Mixup on OOD detection for high-resolution datasets, such as those in [4], especially when using 224x224 ImageNet as the ID dataset.


[1] INTRA-CLASS MIXUP FOR OUT-OF-DISTRIBUTION DETECTION

[2] CutMix: Regularization Strategy to Train Strong Classifiers with Localizable Features

[3] MixOOD: Improving Out-of-distribution Detection with Enhanced Data Mixup

[4] OpenOOD: Benchmarking Generalized

**Questions:**

Please see Weaknesses above.

---

> ### Author Response · Authors · 2023-11-19
>
> We appreciate your thorough and insightful reviews and will address your concerns one by one.
> # W1. Novelty Concerns.
> **R1.** Our contribution goes beyond employing mixup to improve OOD detection from the following perspectives.
> (1) We first present a theoretical and empirical understanding demonstrating a more diverse set of auxiliary outliers is critical in improving OOD detection by lowering the upper bound of OOD detection error. Our findings could offer fresh theoretical insights for future OOD detection advancements.
> (2) Building on the proposed theory, we develop a simple yet effective method by leveraging mixup to enrich outlier diversity with corresponding theoretical evidence. Note that unlike the previous mixup-based methods, our core motivation for introducing mixup is to improve the diversity of the auxiliary outlier dataset instead of designing novel mixup strategy.
> (3) Unlike the insightful methods proposed in [1, 2, 3] by applying mixup to ood detection, our main contribution is to explore a more diverse auxiliary outlier space. This focus sets our research different from existing ones.
>
> # W2. The effectiveness of the modification to mixup from theoretical and experimental perspectives.
> **R2.** **Theoretical Perspective:** As the potential space for auxiliary outliers data can be excessively large, the majority of outliers uninformative for model regularization, which may lead to inefficient and insufficient exploration and exploitation of these outliers [4][5]. To mitigate this, we strategically select the mixed outlier samples that are closest to the in-distribution boundary. These samples are more likely to be misclassified as in-distribution (ID) data and need to receive more attention. In our theory, mixup can generate new outliers with semantics different from the original data, and this selection strategy can select the newly generated outliers that are easy to be misclassified by the model for training.
> **Empirical Validation:** Ablation experiments were performed on this module in Table 2. We present the results here:
> For cifar10 dataset:
>
> |method |FPR $(\downarrow)$|AUROC$(\uparrow)$|AUPR$(\uparrow)$|ID-ACC|
> |-|-|-|-|-|
> |only mixup outliers| 3.28|99.23 |**99.45** |94.30 |
> |mixup outliers + greedy sampling |**2.18** | **99.43** | 99.01|94.32 |
>
> For cifar100 dataset:
> |method |FPR $(\downarrow)$|AUROC$(\uparrow)$|AUPR$(\uparrow)$|ID-ACC|
> |-|-|-|-|-|
> |only mixup outliers| 13.33|97.35 |**98.66** | 74.28|
> |mixup outliers + greedy sampling | **10.37** |**98.03** |98.63 | 74.26|
>
> These results clearly illustrate that selecting Mixed outliers based on their out-of-distribution (OOD) score consistently enhances OOD detection performance compared to using only Mixup. This underlines the significant benefits of our proposed modification in improving the effectiveness of OOD detection.
>
> # W3. Concerns about experimental settings on auxiliary datasets.
> **R3.** Thank you for the detailed and interesting comments. We would address this concern in the following ways:
> (1) In practical applications, access to a more diverse auxiliary dataset is often possible, such as crawling data from the Internet. This approach allows to train the model with a more diverse auxiliary outlier dataset. Consequently, this setting has been adopted by many related studies within the field.
> (2) While ImageNet-RC offers greater diversity compared to datasets like CIFAR-10/100, it still does not fully represent the vast and complex nature of true open-world out-of-distribution (OOD) data. This limitation highlights the need for even more diverse datasets to truly model the real-world data.
> (3) We also experimentally confirm that our method is still effective even when the diversity of auxiliary outliers is not significantly higher than ID data. In the experiment "Q2 Reliability", we tested the effectiveness of our method under the outliers dataset with different diversity. As can be seen from Figure 2, on the cifar-100 dataset, when we restrict the diversity of the outliers dataset (using only 100 classes of the ImagenetRC dataset as outliers, the diversity of the outliers dataset is comparable to that of the ID dataset), the OOD detectors are still able to benefit from outliers. When our method is adopted to improve the diversity of outliers, the performance of the model is significantly improved (The FPR95 metric significantly decreased from 26% to 20%.), which also confirms the effectiveness of our method and theory.
>
> [1] Intra-class Mixup for Out-of-Distribution Detection
>
> [2] CutMix: Regularization Strategy to Train Strong Classifiers with Localizable Features
>
> [3] MixOOD: Improving Out-of-distribution Detection with Enhanced Data Mixup
>
> [4] POEM: Out-of-Distribution Detection with Posterior Sampling
>
> [5] ATOM: Robustifying Out-of-distribution Detection Using Outlier Mining

---

> > ### Author Response · Authors · 2023-11-19
> >
> > # W4.  Experiments on high-resolution datasets.
> > **R4.** We appreciate the suggestion to evaluate our approach on larger-scale benchmarks and provide the corresponding experiments under the setting in [6]. Our experimental setup involves utilizing 200 classes from ImageNet-1k (224x224 resolution) as the in-distribution (ID) class and an additional 800 classes as outliers. We conduct tests across five different out-of-distribution (OOD) test sets. Specifically, SSB-hard and NINCO are included as part of the near-OOD group for ImageNet-1k, while iNaturalist, Textures, and OpenImage-O are classified under the far-OOD group. The experimental results are as follows:
> >
> > |  FPR95 ($\downarrow$) / AUROC ($\uparrow$)  | NINCO | SSB_hard | Textures | iNaturalist | OpenImage-O | average |
> > |--- | -----  | -------- | ---      | ----------- | --------- | ------- |
> > |MSP | 78.90 / 80.24 | 76.25 / 81.11 | 67.64 / 81.97 | 76.12 / 82.77 | 78.19 / 80.90 | 75.42 / 81.40 |
> > | energy (w.o. outliers) | 73.56 / 84.12 | 68.86 / 84.91 | 52.55 / 86.78 | 67.68 / 87.66 | 72.76 / 84.18 | 67.08 / 85.53 |
> > | k+1 | 77.48 / 82.31 | 73.14 / 84.61 | 71.95 / 84.47 | 91.49 / 78.87 | 75.64 / 83.59 | 77.94 / 82.77 |
> > | OE | 67.78 / 84.85 | 65.07 / 85.81 | 51.31 / 88.92 | 65.72 / 86.64 | 62.75 / 86.71 | 62.53 / 86.59 |
> > | promix (k+1) | 50.37 / 90.42 | 43.99 / 91.76 | 55.62 / 88.91 | 45.30 / 92.33 | 40.31 / 92.80 | 47.12 / 91.24 |
> > | promix (OE) | **42.80 / 90.82** | **36.75 / 92.19** | **35.41 / 92.12** | **25.86 / 95.12** | **33.48 / 93.43** | **34.86 / 92.74** |
> >
> > Among them, the MSP and energy methods were trained without using outliers, while the k+1 and OE methods were trained with outliers. To validate the versatility of our method, we adapted it to the OOD scores of k+1 and OE, denoted as promix(k+1) and promix(OE), respectively. The experimental results show that our method is quite effective on high-resolution and large-scale  datasets both on near OOD detection and far OOD detection.
> >
> > [6] OpenOOD v1.5: Enhanced Benchmark for Out-of-Distribution Detection

---

> ### Comment · Reviewer_pfu4 · 2023-11-23
> **Further comments**
>
> The authors' responses address most of my concerns—however, the concern regarding the auxiliary dataset remains. Thus, I would like to stand by my original score. However, I would like to note that I will not fight for a rejection.

---

> > ### Author Response · Authors · 2023-11-23
> >
> > Thank you for providing your feedback. We would like to address the misunderstanding about auxiliary outlier dataset.
> >
> > Our method follows the standard settings of using outlier datasets for OOD detection in previous methods [1, 2, 3, 4, 5, 6, 7,8 ,9 ,10] published at top-tier conferences.
> >
> > [1] Deep anomaly detection with outlier exposure. (ICLR, 2018)
> >
> > [2] Unsupervised out-of-distribution detection by maximum classifier discrepancy. (ICCV, 2019)
> >
> > [3] Energy-based Out-of-distribution Detection. (NeurIPS, 2020)
> >
> > [4] Semantically coherent out-of-distribution detection. (ICCV, 2021)
> >
> > [5] Atom: Robustifying out-of-distribution detection using outlier mining. (ECML PKDD, 2021)
> >
> > [6] POEM: Out-of-Distribution Detection with Posterior Sampling. (ICML, 2022)
> >
> > [7] Self-supervised learning for generalizable out-of-distribution detection. (AAAI, 2022)
> >
> > [8] Training ood detectors in their natural habitats. (ICML, 2022)
> >
> > [9] Unknown-Aware Object Detection: Learning What You Don't Know From Videos in the Wild. (CVPR, 2022)
> >
> > [10] Out-Of-Distribution Detection with Implicit Outlier Transformation. (ICLR, 2023)
> >
> > We hope this clarification provides a clearer perspective on our settings.

---

### Meta-Review · Area_Chair_y61P · 2023-12-05

**Metareview:**

This work aims to address the challenges in OOD detection, particularly the limitation of current detectors to generalize from the distribution of auxiliary outliers. The authors introduce Provable Mixup Outlier (ProMix), using Mixup to increase the diversity of auxiliary outliers, with insightful theoretical analysis, leading to enhanced OOD detection. Evaluations on benchmarks like CIFAR-10 and CIFAR-100 indicate improved performance over existing techniques. Specifically, the strength of this paper includes several aspects. 1) The paper is well-structured, making it reader-friendly and easy to follow. 2) The theoretical analysis constructed for Mixup is both enlightening and insightful. 3) Extensive ablation studies were conducted, showcasing various experiment setups and results.

However, there are several points to be further improved. For example, the auxiliary dataset used for training is a downsampled version of ImageNet. Compared to the utilized ID datasets CIFAR-10 and CIFAR-100, its diversity seems significantly higher. Therefore, the OOD detection performance gains achieved with such an auxiliary dataset might not be wholly convincing, even though the methods authors compared against also adopt this approach. Moreover, why can mixup create diverse outlier examples? It will not change the data distribution much since it is only doing the interpolation work. What if the auxiliary outlier set is very constrained in the convex high-dimensional input space, would mixup increase the diversity of the outlier set significantly? If possible, the authors are encouraged to provide empirical evidence on large-scale benchmarks. The current setting does not seem to suggest the diversity will create a performance improvement significantly. Therefore, this paper cannot be accepted at ICLR this time, but the enhanced version is highly encouraged to submit other top-tier venues.

**Justification For Why Not Higher Score:**

However, there are several points to be further improved. For example, the auxiliary dataset used for training is a downsampled version of ImageNet. Compared to the utilized ID datasets CIFAR-10 and CIFAR-100, its diversity seems significantly higher. Therefore, the OOD detection performance gains achieved with such an auxiliary dataset might not be wholly convincing, even though the methods authors compared against also adopt this approach. Moreover, why can mixup create diverse outlier examples? It will not change the data distribution much since it is only doing the interpolation work. What if the auxiliary outlier set is very constrained in the convex high-dimensional input space, would mixup increase the diversity of the outlier set significantly? If possible, the authors are encouraged to provide empirical evidence on large-scale benchmarks. The current setting does not seem to suggest the diversity will create a performance improvement significantly. Therefore, this paper cannot be accepted at ICLR this time, but the enhanced version is highly encouraged to submit other top-tier venues.

**Justification For Why Not Lower Score:**

However, there are several points to be further improved. For example, the auxiliary dataset used for training is a downsampled version of ImageNet. Compared to the utilized ID datasets CIFAR-10 and CIFAR-100, its diversity seems significantly higher. Therefore, the OOD detection performance gains achieved with such an auxiliary dataset might not be wholly convincing, even though the methods authors compared against also adopt this approach. Moreover, why can mixup create diverse outlier examples? It will not change the data distribution much since it is only doing the interpolation work. What if the auxiliary outlier set is very constrained in the convex high-dimensional input space, would mixup increase the diversity of the outlier set significantly? If possible, the authors are encouraged to provide empirical evidence on large-scale benchmarks. The current setting does not seem to suggest the diversity will create a performance improvement significantly. Therefore, this paper cannot be accepted at ICLR this time, but the enhanced version is highly encouraged to submit other top-tier venues.

---

### Decision · Program_Chairs · 2024-01-16

Reject